# Threads of memory: Reviving the ornament of a dead child at the Neolithic village of Ba'ja (Jordan)

Hala Alarashi[1,2]*, Marion Benz[3], Julia Gresky[4], Alice Burkhardt[5], Andrea Fischer[5], Lionel Gourichon[2], Melissa Gerlitzki[6], Martin Manfred[6], Jorune Sakalauskaite[7,8], Beatrice Demarchi[9], Meaghan Mackie[7], Matthew Collins[10], Carlos P. Odriozola[11,12], José Ángel Garrido Cordero[12], Miguel Ángel Avilés[13], Luisa Vigorelli[14,15], Alessandro Re[15], Hans Georg K. Gebel[3,16]

1 IMF-CSIC, Barcelona, Spain, 2 Université Côte d'Azur, CNRS, CEPAM, Nice, France, 3 Institute of Near Eastern Archaeology, Free University, Berlin, Germany, 4 German Archaeological Institute, Berlin, Germany, 5 Department of Conservation–Art History, Stuttgart State Academy of Art and Design, Stuttgart, Germany, 6 Landesamt für Geologie, Rohstoffe und Bergbau (LGRB) im Regierungspräsidium Freiburg, Freiburg, Germany, 7 Section for GeoBiology, GLOBE Institute, Faculty of Health and Medical Science, University of Copenhagen, Copenhagen, Denmark, 8 Institute of Bioscience, Life Sciences Centre, Vilnius University, Vilnius, Lithuania, 9 ArchaeoBiomics, Department of Life Sciences and Systems Biology, University of Turin, Turin, Italy, 10 McDonald Institute for Archaeological Research, University of Cambridge, Cambridge, United Kingdom, 11 Departamento de Prehistoria y Arqueología, Universidad de Sevilla, Seville, Spain, 12 UNIARQ, Centro de Arqueologia da Universidade de Lisboa, Lisbon, Portugal, 13 Instituto de Ciencia de Materiales de Sevilla, Universidad de Sevilla- CSIC, Seville, Spain, 14 Electronics and Telecomunication Department, Polytechnic of Torino, Turin, Italy, 15 Physics Department, University of Torino and INFN, Turin Section, Turin, Italy, 16 ex oriente e.V., Berlin, Germany

* hala.alarashi@imf.csic.es, alarashi.hala@gmail.com

**Data Availability Statement:** All relevant data are within the paper and its Supporting Information files.

## Abstract

In 2018, a well-constructed cist-type grave was discovered at Ba'ja, a Neolithic village (7,400–6,800 BCE) in Southern Jordan. Underneath multiple grave layers, an 8-year-old child was buried in a fetal position. Over 2,500 beads were found on the chest and neck, along with a double perforated stone pendant and a delicately engraved mother-of-pearl ring discovered among the concentration of beads. The first was found behind the neck, and the second on the chest. The meticulous documentation of the bead distribution indicated that the assemblage was a composite ornament that had gradually collapsed, partly due to the burying position. Our aim was to challenge time degradation and to reimagine the initial composition in order to best explore the significance of this symbolic category of material culture, not as mere group of beads, but as an ornamental creation with further aesthetic, artisanal and socioeconomic implications. The reconstruction results exceeded our expectations as it revealed an imposing multi-row necklace of complex structure and attractive design. Through multiple lines of evidence, we suggest that the necklace was created at Ba'ja, although significant parts of beads were made from exotic shells and stones, including fossil amber, an unprecedented material never attested before for this period. The retrieval of such an ornament from life and its attribution to a young dead child highlights the significant social status of this individual. Beyond the symbolic functions related to identity, the necklace is believed to have played a key role in performing the inhumation rituals,

**Funding:** - ArchaeologyHub.CSIC 2022 Internal Research Grant (HA) https://archaeologyhub.csic. es/ - H2020 Marie Sklodowska-Curie Actions, grant number 846097 (HA) https://cordis.europa.eu/ project/id/846097 - German Research Foundation (BO 1599/14-1; BO 1599/16-1) (MB, HGG) https:// www.dfg.de/en/ - Franz-and Eva Rutzen Stiftung Foundation (MB) https://www.deutsches-stiftungszentrum.de/stiftungen/franz-und-eva-rutzen-stiftung - Junta de Andalucía (Consejería de Economía, Conocimiento, Empresas y Universidad), under contract P20_01080 (CPO) https://www.juntadeandalucia.es/organismos/ universidadinvestigacioneinnovacion.html The funders had no role in study design, data collection and analysis, decision to publish, or preparation of the manuscript.

**Competing interests:** The authors have declared that no competing interests exist.

understood as a public event gathering families, relatives, and people from other villages. In this sense, the necklace is not seen as belonging completely to the realm of death but rather to the world of the living, materializing a collective memory and shared moments of emotions and social cohesion.

## Introduction

Adornments are powerful cultural symbols that communicate personality, values, merits, intentions, and artistic tastes through their display on the body [1–7]. Few things are as visually compelling and impactful as the act of marking, dressing, and adorning oneself when it comes to asserting ideas, identities, or positions [5, 8, 9]. However, expressing individual identity contradicts the definition of the individual as an "*active subject establishing dialectical relations with its social and natural environment*" [10] (p. 137), and as a social self with shared experiences, memories, and stories [11]. This paradoxical logic is based on the idea that adornments require a pre-existing cultural code to convey their meaning. This code consists of implicit rules and conventions that are understood by the people who create, own, and display them. In this sense, conventionalized adornments transcend individual identity to represent a group of individuals [12], becoming thereby as complex as the concept of identity itself insofar it is "*at once both imposed by others and self-imposed*" [5] (p. 210). Hence, body adornments, which play an active role in shaping and enhancing the individual character [13, 14], express inexorably a shared sociocultural identity. Nevertheless, how should one consider outstanding and unique ornaments, innovative creations that do not resemble any of what have existed before? Wouldn't their mere recognition as different justification for inclusion in a code of classification; a first step towards normalization? Although the answers to these questions are beyond the scope of this article, they do highlight the complexity and ambiguity of the symbolic functions of ornaments. On the other hand, these aforementioned approaches emphasize the potential that body ornaments possesses for providing insights into the prehistoric communities. Indeed, body adornments within prehistoric societies are no exception to rules and conventions. The use of specific materials and colors to create specific types is a clear indication of how past societies applied and repeated cultural standards, including imitation practices which underline the importance of certain forms [15–18] or raw materials [19, 20].

Determining the extent to which an ornamental tradition was conventionalized based on factors such as gender, age, socio-cultural or ethnic background is a challenging task. In fact, investigating these aspects of past preliterate societies largely depends on the funerary record. It also requires identifying and comprehending the material culture that arose from the interplay between the living and the deceased. As some studies have warned, e.g. [21, 22], burial rites can be influenced by socioeconomic and political factors, thus diverging from the realities of the living. However, graves serve as a crucial data source for exploring the dialectic relation between individuals and their societies, as well as the broader social structure of prehistoric groups. Indeed, funerary practices provide insights into the interconnectivity of individuals at multiple household, village, and regional scales [23], even when these connections are not readily apparent through material culture. In turn, materials and artifacts uncovered in graves, which are often of a higher quality than those found in non-funerary contexts, enrich our knowledge of the economic, technological and artisanal abilities of these societies. They also provide insight into the breadth and intensity of their exchange networks and cultural interactions.

In the Near East, funerary rituals and treatment of the dead became clearly identifiable and consistent as early as the Epi-Paleolithic pre-Natufian [24, 25] and continued to be evident during the Late Epi-Paleolithic, Natufian culture [26–28]. The complexity increased during the Early Neolithic, reaching a culminant point during the Pre-Pottery Neolithic B as primary burials were opened and re-opened, human remains were manipulated and displaced [29, 30], human faces were recreated through skull plastering and paintings [31–33], and in specific contexts, even cremations were practiced [34]. Considering this complexity in context of the establishment of a farming lifestyle, the substantial increase in population size, and the unprecedented intensification of human relationships at various geographical scales [35–37] have led researchers to reflect and model the social organization while exploring patterns of differentiation [38–41], or looking for signs of egalitarian social structures [42–44]. Surprisingly, tackling changes in the social structures during the Neolithization process in the Levant is rarely approached from the individuals' perspective, as active subjects within their socioeconomic and cultural *milieus*. The extraordinary treatments given to certain individuals upon their death can be indeed evaluated and compared to detect similarities or patterns of differentiations [6, 45]. However, when it comes to performing rituals, it is important to consider the public dimension [38, 46–48] and the collectivity, especially in the realm of funerary practices where the potential of reality distortion is high due to emotional charge [49, 50], to social or political objectives that need to be achieved [11, 22, 51, 52], or even for the purpose of consolidating and extending socioeconomic networks [53]. The size of the audience gathered for the event of death increases the intensity of these collective moments. The burial construction and how the body was staged inside are actions that encompass ritual performances in which funerary goods play a crucial role. Indeed, for most of the cultures of the world, material culture, in particular those related to face and body adornments, is fundamental in the performance of rituals [14, 31, 47, 54, 55]. In this sense, investigating individuals through death rituals and funerary goods is not only about identifying their qualities, status, and identities. It is also a reflection of the impact of death on the living, showcasing the intensity of their expressions and the extent to which they invest in materializing their thoughts and emotions. Drawing upon these approaches, this study explores the significance of an exceptional ornament discovered in grave CG7 at Ba'ja, a Late Pre-Pottery Neolithic B (LPPNB) village in Southern Levant (Fig 1). Through the use of multiple lines of evidence, our aim is to assist, on the one hand, understanding the ornamental practices at Ba'ja and their implication in death rituals, and on the other, to evaluate to what extent body ornaments contributed to boosting the technological, economic and social systems of the farming communities at the end of the PPNB in the Levant.

Grave CG7 was discovered in the summer 2018, beneath the floor of room CR 36.1 at Ba'ja (Fig 2). Its excavation has revealed the skeletal remains of an approximately 8-year-old child along with concentrations of beads varying in types, sizes, and colors were found on the chest area and around the neck. Despite the meticulous excavation and careful documentation of the position of the beads (S1 Appendix), organization patterns were only partially detected. During the excavation, it became however obvious that these beads were part of an ornament that has collapsed and disorganized gradually after the decomposition of the body and the associated organic materials (strings, tissues, etc.). In this paper, we demonstrate how it is possible to overcome the effects of time degradation and excavation conditions to unearth and reconstruct a multimillennial creation that have never been attested before in the Levant. The analyses were conducted at two levels. We firstly addressed the morpho-typological diversity of beads, the colors, the raw materials used, the manufacturing schemes, and the patterns of intensity of use, thus providing biographic information about the objects. Then, we used these results, along with data collected during the excavation and a series of morphometric and mathematical estimations, to model the structure of the ornament and the combination of

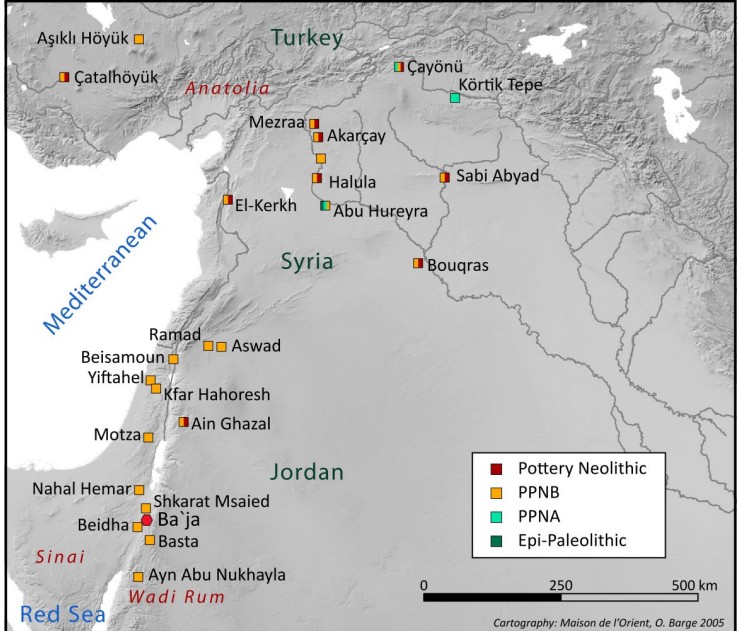

**Fig 1. Map showing the location of Ba'ja and other PPN sites.** Most of the sites are mentioned in the text or concerned by the cited bibliography.

beads according to the most plausible, objective, and simplest scenario possible. The result of the reconstruction exceeded our expectations. The ornament revealed a complex, imposing and very attractive composition that is worth presenting not only to the scientific community but also to the public. As such, the ornament was physically reconstructed and now on display as part of the permanent exhibition at the new Petra Museum in Southern Jordan since October 2021.

## The village of Ba'ja

The site of Ba'ja is located in the Greater Petra Area in Southern Jordan, not far from the well-known historical city of Petra. It is one of the most important and well-studied LPPNB villages in the Southern Levant. Since 1997, archaeological fieldwork has exhumed in several areas of the site, massive, deep, and complex buildings. These architectural complexes consist of multi-cellular ground plans topped by a main store and equipped with domestic facilities. Such complexes were considered as household units [56] in which the bodies of deceased were interred as single, double or collective/multiple burials [57] (Fig 2). The intensive fieldwork has therefore produced exceptional data regarding the ritual and symbolic behavior of the inhabitants of Ba'ja vis-à-vis death.

The main occupation of the settlement is dating to the LPPNB [58], most probably between *c*. 7,400–6,600 cal BCE as indicated by the new radiocarbon dating [45, 59]. Despite its small size of only about 1,5 ha, the density of buildings, and the architectural and organizational features of the village among other characteristics led it to be associated with the "mega-sites" phenomenon that flourished at the end of the PPNB [60]. Yet, unlike other contemporaneous villages, Ba'ja was built in a territorially demanding topographic setting on an intra-mountain plateau (∼1180 m a.s.l.) bordered by impressive gorges and vertical rock formations (Fig 2). Even more intriguing is that the access to the village is through a tortuous ascending gorge to the west, Siq al-Ba'ja [61].

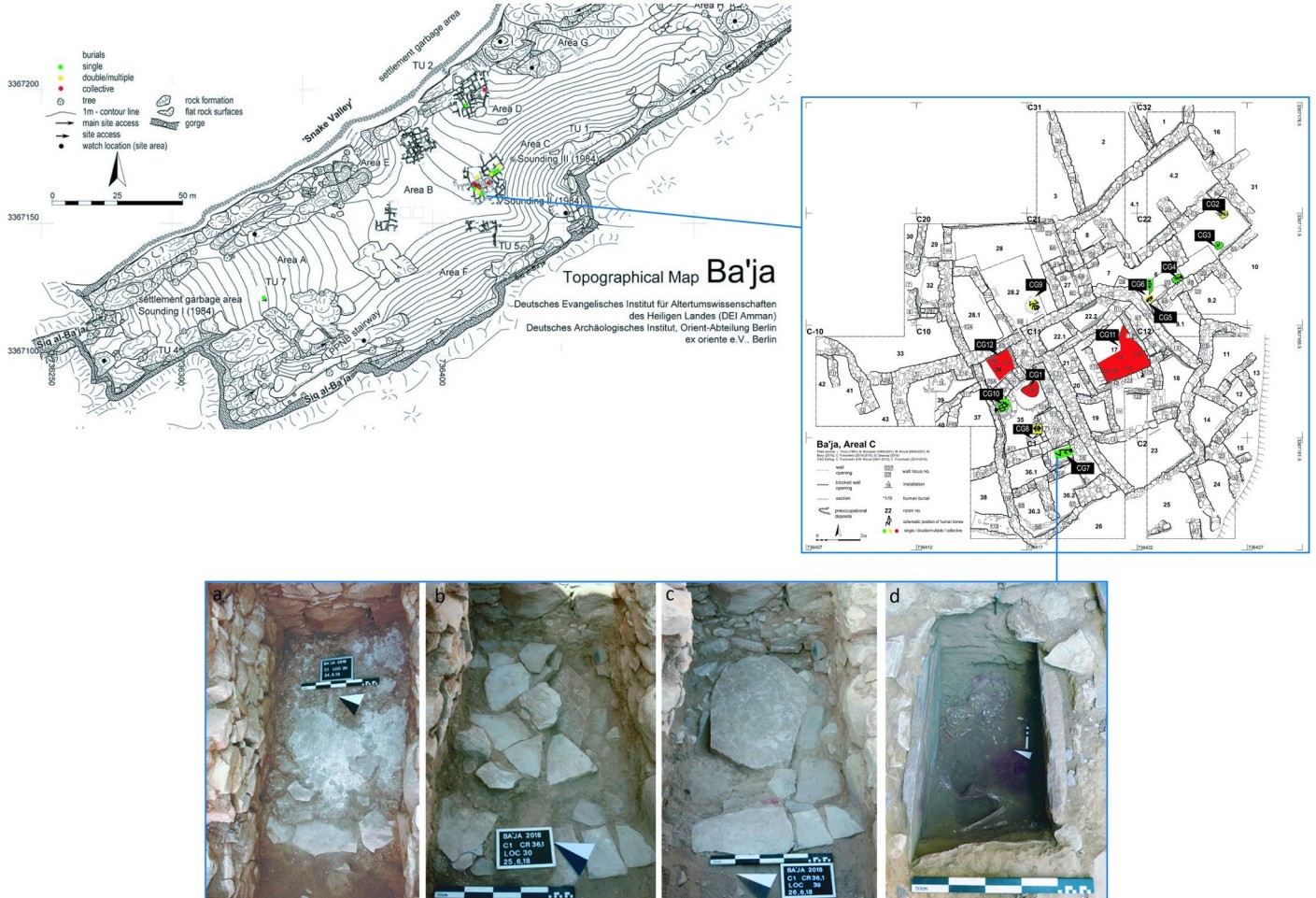

**Fig 2. Location of the cist-type grave CG7 at Ba'ja, Area C, and its progressive excavation stages.** Topographic plan of Ba'ja, the architectural plan of Area C and the location of grave CG7 in Room 36.1. A-C) The grave sealed with different layers of sandstone slabs. D) The child's skeleton in a crouched position resting on her/his left side. Photos: A-C: M. Benz; D: H. Alarashi, Ba'ja N.P.

While most of the deceased were likely buried elsewhere, specific individuals were selected for burial beneath the house floors [57]. Between 2001 and 2021, 15 burials were discovered, with a notably high number of infant burials [45]. Subadults were buried in various types of burials, including single, double, multiple, and collective, while adults were primarily interred in collective burials with few to no grave goods. However, there were two exceptions: a highly elaborate burial of a young man [59], which resembled the construction of grave CG7, and a so-called "trash burial," the only burial discovered outdoors so far.

## The discovery of the child's ornament

Grave CG7, excavated to the natural soil, is a complex cist-type structure with several covering layers (Fig 2A–2C). A total of ten events, including the construction of the grave, the interment of the child and the deposition of the funerary goods were distinguished (for a complete description of these events, and the plan of the burial refer to [58] figs 4, 5, 6 and table 1). A child, presumably a girl (based on the presence of mental protuberance on the mandible) [45] around eight years of age, was buried on the left side in a fetal position (Fig 2D). Due to the

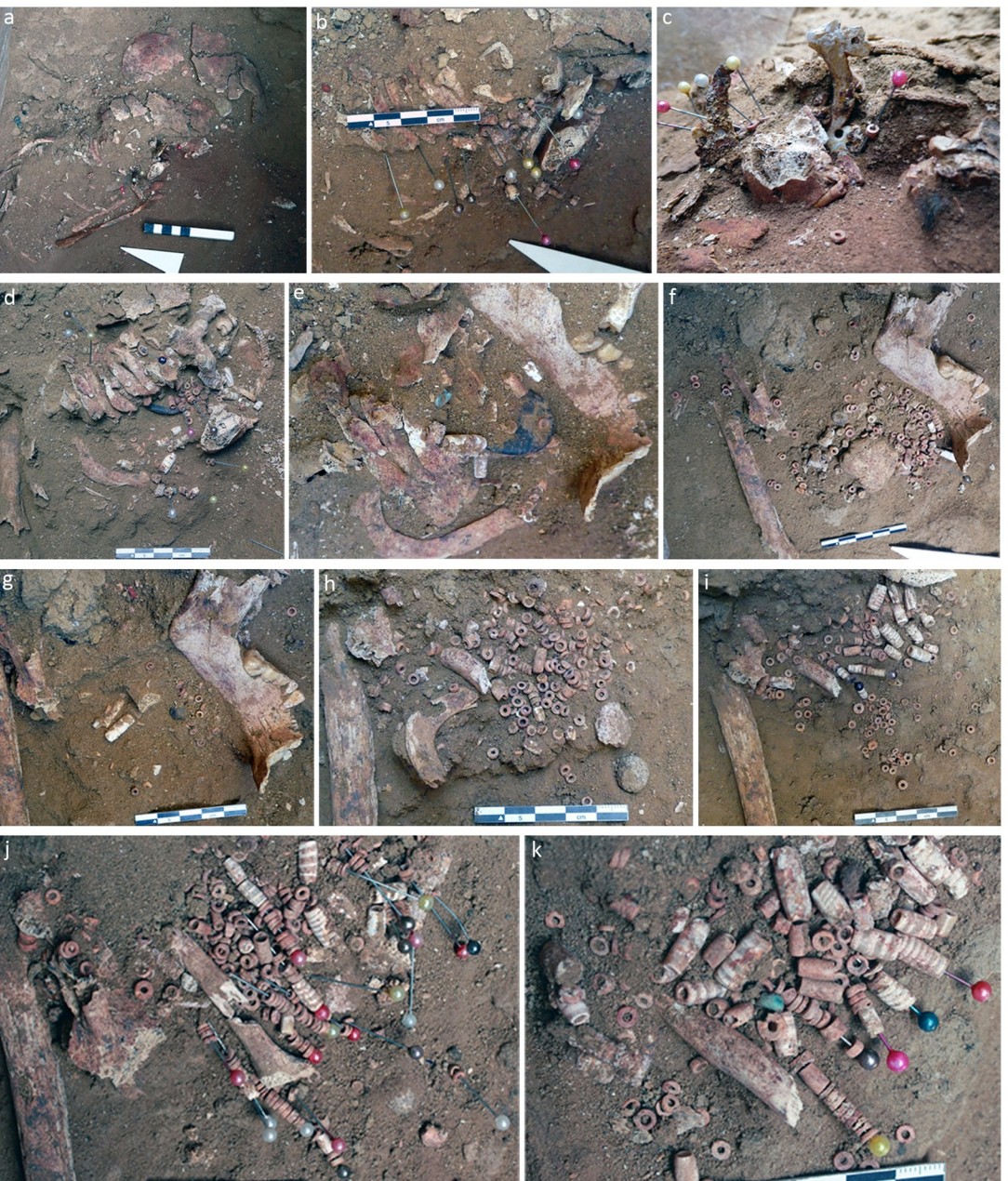

**Fig 3. Field pictures showing the densities and areas of distribution of the beads.** A) A general view of the damaged skull and upper part of the child's skeleton with few scattered beads fixed with pins. B) Picture showing the broken engraved ring (to the right) and its connection with disc beads. C) The ring pictured as discovered in a vertical position laying on its edge, with disc beads still stuck to the perforations decorating its surface. D) Increased density of the beads after the removal of the ring and the appearance of a black stone element below the cervical vertebra. E) Beads stuck to the perforation of the black element and appearance of an additional turquoise bead (note the fragmentation and the displacement of the mandible). F) The manubrium bone and a spherical black bead on its upper left. G) Another spherical black bead appeared just after the removal of the manubrium, placed at its upper right. H-K) Increased density of beads alignments in the area between the left clavicula and the left part of the mandible (removed). Photos: A, J, K: M. Benz; B-I: H. Alarashi, Ba'ja N.P.

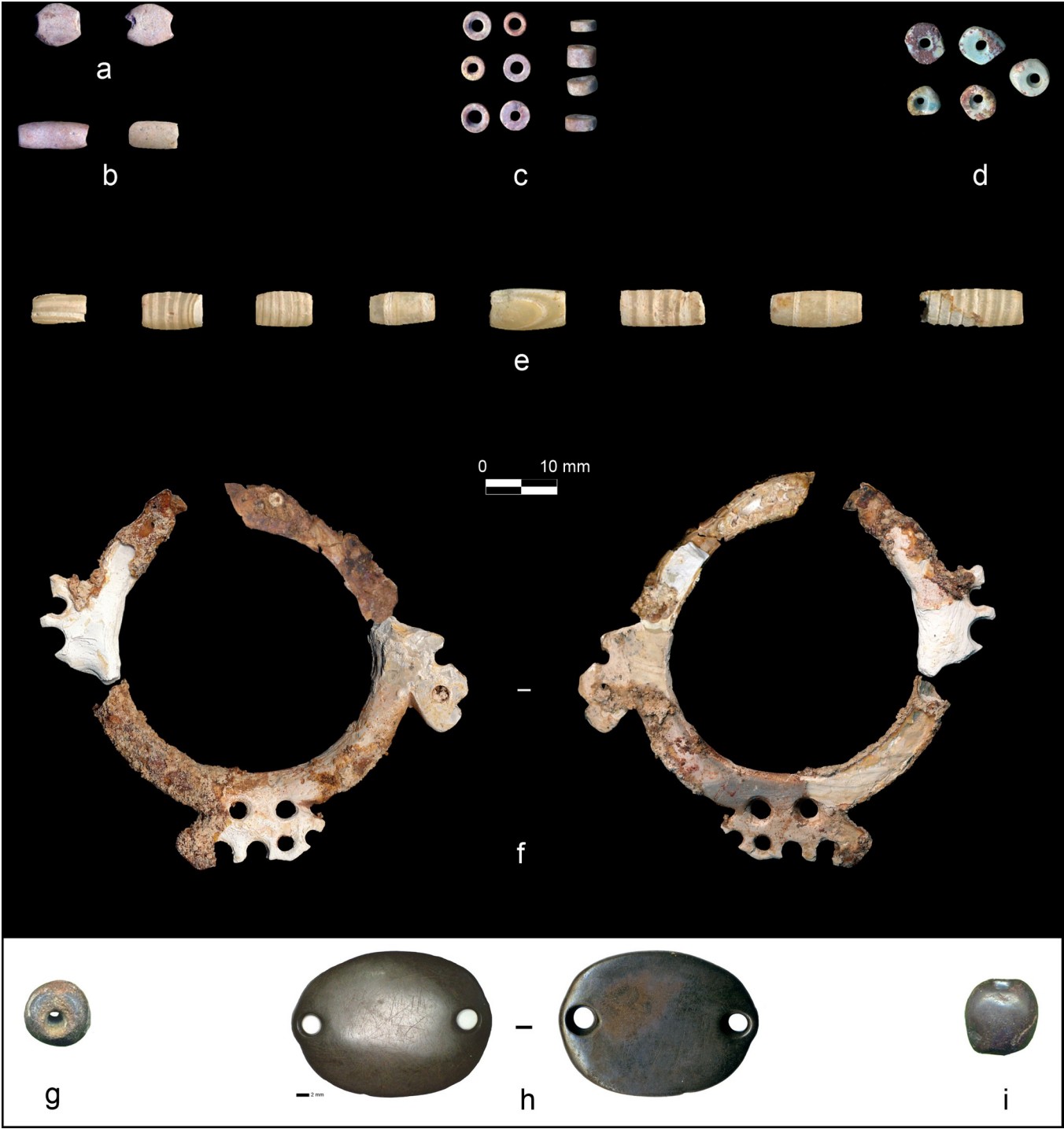

**Fig 4. Diversity of items compositing the ornamental assemblage of CG7.** A) Flat beads, B) Tubular cylindrical beads. C-D) Disc beads. E) Tubular shell beads. F) Mother-of-pearl multi-perforated and engraved ring viewed from both faces. G and I) Compact sub-spherical stone beads. H) Double perforated pendant viewed from both faces. Photos: A-E, G-H: H. Alarashi, F: A. Burkhardt, Ba'ja N.P.

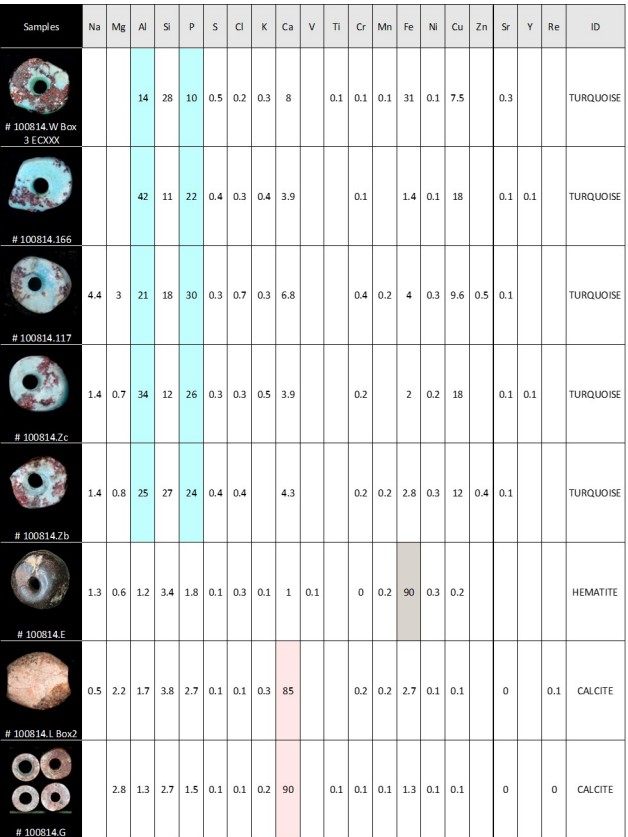

**Fig 5. The results of XRF analyses of a selection of beads found in CG7.** Amounts of chemical elements composing the samples and ID of materials. Highlighted elements are typical of the identified stones. Photo: H. Alarashi, Ba'ja N.P.

poor preservation of bones, the genetic and isotopic analyses failed in providing information regarding the child's biological identity, dietary habits, or health status. Consequently, we focused on the associated material culture.

As mentioned before, the skeleton is very poorly preserved. The right arm was missing, and parts of the skull, mandible, pelvis and feet were severely damaged by post-depositional factors such as the heavy and layers covering the body. However, the preserved skeletal remains were well articulated. Most of them were stained red on their outer surface suggesting that the child either had clothes, or the skin was stained red. The skull was turned on its face. The meticulous excavation revealed more than 2500 beads mostly concentrated in the areas corresponding to the chest and the neck of the child (Fig 3A, 3B and 3D–3K). Observations made during the excavation, drawings of the layers of beads presenting their distribution and relations to the bones, along with elements of comprehension of the discovery are detailed in S1 Appendix. Several series of aligned beads were found below the left side of the neck, clavicula and the upper ribs, indicating that the beads were gathered and organized into several rows (Fig 3D–3K). Due to gravity, the right side of the ornament had slipped into the area of the neck and the left side of the chest. Some beads were found scattered and isolated due to the heavy damage the right side of the skeleton. The possibility that the beads were simply and randomly deposited on the upper parts of the body can definitely be excluded as series of beads showed repeated patterns of organization and combinations (e.g., after around 10-disc beads, two tubular beads can be found Fig 3J and 3K). These organized portions were observed mainly

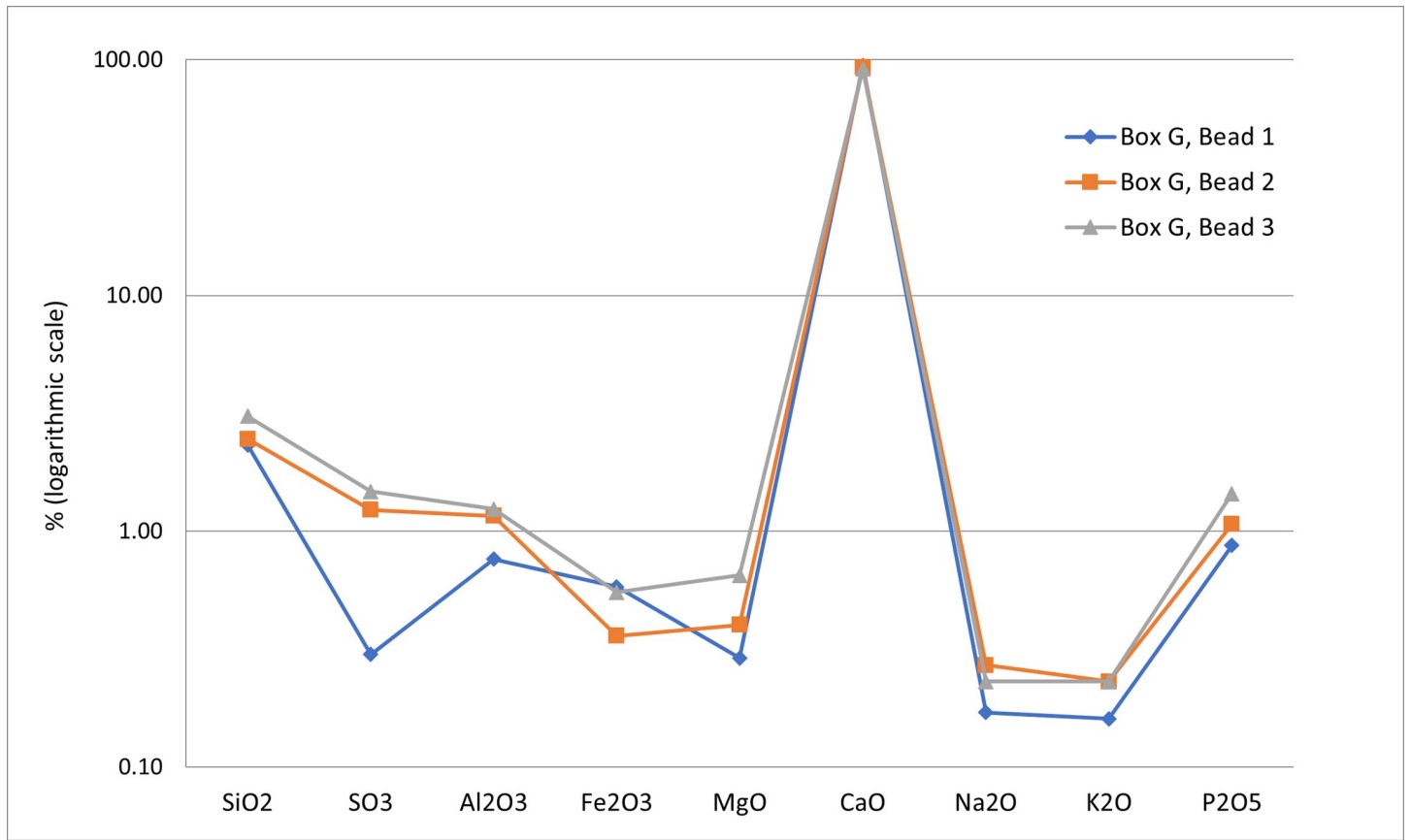

**Fig 6. Analyses of thin sections the red stone disc beads from grave CG7.** Percentage of the chemical elements based on XRF measurements. Graph: M. Benz, Ba'ja N.P.

associated with the left skeletal remains. Such bead arrangements were impossible without a hanging/fastening system. Several questions were formulated straightway during the excavation: were these beads part of the same composition, and if yes, what was the type of ornament in which they were integrated: a necklace, a plastron, a decorated garment?

**Table 1. Diversity of raw materials in relation to the typological families of the items discovered in CG7.**

| Materials | | Beads | | | | Double P. pendant | Flat ring | Total |
|---|---|---|---|---|---|---|---|---|
| | | Disc | Tubular | Flat | Spheric | | | |
| Minerals | Red calcite | 2264* | 66 | 7 | | | | 2337 |
| | Turquoise | 5 | | | | | | 5 |
| | Hematite | | | | 2 | 1 | | 3 |
| Marine shells | *Pinctada* sp. | | | | | | 1 | 1 |
| | *Tridacna* sp. | | 232 | | | | | 232 |
| | Dentalium | | 1 | | | | | 1 |
| | *Conus* sp. | 4 | | | | | | 4 |
| Amber | Amber | | 2 | | | | | 2 |
| Total | | 2273 | 301 | 7 | 2 | 1 | 1 | 2585 |

Double P. pendant = Double Perforated pendant.

*To this should be added over 50 fragments of disc beads (N of the broken beads was not possible to estimate).

Several types of beads made from different colorful materials were distributed between two large objects, a decorated multiperforated mother-of-pearl ring (Fig 4F) and a double perforated pendant (Fig 4H) found behind the neck (Fig 3D and 3E). Several beads were still connected to the ring (Fig 3C) and the pendant (Fig 3E), stuck to some perforations.

The complexity of the arrangement of the beads was perceived as the density of alignments and types increased exponentially. As explained earlier, the highest density in concentration concerns the lowest layers, those corresponding to the left side of the child, reaching down to the bottommost surface of the burial on which the body was laid. The right side of the child corresponds to the upper layers where the beads were scattered and found in low density. This configuration suggests that the right side of the ornament has gradually collapsed after the decomposition of the body and of the organic materials used to gather and fix the beads. The idea of reconstructing the initial composition of the ornament in grave CG7 appeared very challenging, nevertheless achievable, given the detailed documentation collected during the excavation. This goal was established taking into account the significance of composite ornaments from various perspectives. Indeed, composite ornaments are much more informative than individual isolated beads, the most frequently found in the archaeological records [62], because information are conveyed through combinations of types, colors, number of rows, or display on specific parts of the body, etc. Moreover, reconstructing the initial configuration provides a rare opportunity to address the complexity of designs, as this endeavor has never been carried out before in the context of farming lifestyle in the Levant. Finally, it allows exploring important concepts such as light, symmetry, volume and verticality, that are rarely considered for this period.

## Methods

At the time of the discovery, many objects presented post-depositional alterations due to calcifications. Several items were agglomerated, preventing their examination or even counting. Moreover, the huge number of items, the narrowness of the burial space to excavate in optimal conditions, the time constraints, led us to set up very specific strategies of material recovery (details in S1 Appendix). The whole assemblage was exported to Europe under a loan permit obtained from the Jordanian authorities for study, restoration and reconstruction (The beads were loaned under the permits numbers 12/5/274 and 12/5/1334). The beads were first sorted into two groups: the "group of Stuttgart" (GS), composed from 888 items (35%) that remained at the State Academy of Art and Design of Stuttgart (Stuttgart, Germany) as they needed cleaning, consolidation and/ or restoration [63]; and the "group of Nice" (GN) totalling of 1697 relatively well-preserved items (65%) on which it was possible to carry out macro- and microscopic diagnoses and morphometrical analyses. The study of this group was carried out at the CEPAM laboratory (CNRS-UMR 7264), University Côte d'Azur (Nice, France).

### Excavation and field documentation

Despite the difficulty of the excavation due to the loose sediments inside the burial, the ornament was gradually unearthed by revealing, layer by layer, the position and organization of the beads associated with the skeletal remains. From the uppermost to the lowest layer, the items discovered were photographed, sketched, and then removed and placed with specific labels into bags. Grave filling was systematically sieved with a 0.5 mm sieve. A detailed description of the system of documentation and the work conditions are provided in S1 Appendix.

## Identification of raw materials

A first classification of the material types was based on visual aspects (color, textures) of the surfaces and on the inner matrix exhibited when some beads were broken. While turquoise and hematite are recognizable due to their specific macroscopic characteristics, the reddish soft, the white with alternative translucent and opaque bands, and the brownish resin-like materials could not be unequivocally classified into stones, shells, or another material categories. Geochemical, structural, and proteomic analyses were then performed to refine preliminary identifications, thus allowing more precise identifications of the provenience. The description of the methods employed for identifying the shell and the resin-based beads are detailed in S2 Appendix, parts I and II.

## Typology and morphometry

Functional types and their shapes were determined based on a classification method established by Alarashi [64] for prehistoric ornamental objects. All the elements were 2D scanned using an Epson Perfection 4490 PHOTO scanner. Only the GN (group of Nice) samples were measured and morphometrically analyzed. Linear measurements were made manually using an electronic calliper, with values filled automatically on an MS Excel spreadsheet. For disc beads, the greatest diameter, the diameter of perforation, and the degrees of roundness and circularities (for details on the last two parameters, *cf*. S2 Appendix, part IV) were calculated through Automatic Pattern Recognition System by applying informatic scripts programmed by the first author using an open-source image processing software (ImageJ Fiji 2.9.0).

The fragile mother-of-pearl ring was measured from its scanned image. The tubular white beads have visible growth lines in the shape of parallel opaque to translucent bands or stripes (Fig 4E). These were described according to the main major axis of the bead (perpendicular, parallel, and oblique). The aim was to examine if there was a correlation between these patterns and the distribution and position of the tubular beads within the ornament.

## Macro- and microscopic observations

Due to its poor state of preservation and fragility [63], we examined the mother-of-pearl ring using a portable digital microscope (Dino-Lite Edge 5m, 20X – 200X) to avoid direct handling as much as possible. The study took place in the laboratory of Stuttgart before the restoration process. The GN objects were examined to determine the types and sections of perforations. Microscopic traces related to the manufacturing and use-wear processes were recorded for a total of 440 items (25.92%), representing the whole diversity of raw materials and types. The first observations were made during the excavation in 2018, using a binocular stereoscope (Euromex SteroBlue 0.7X–45X). A macroscope (Leica Z16 APO 0.7X–90X) equipped with a digital camera was then employed at the CEPAM laboratory in Nice.

## Micro-CT analysis

In order to best document the drilling techniques of the tubular beads, Micro-CT scan (details on the method and instrument provided in S2 Appendix, part III) was conducted for two shell beads. The samples were scanned longitudinally and transversally. The first axis allows the visualization of the section of the bidirectional drillings, while the second documents the regularity of the circular shape of the drillings at different distances between the surface and the junction area (where the bidirectional drillings meet).

## Reconstruction of the ornament

Most of the beads found on the first excavated upper layers, those corresponding to the right side of the body were disturbed due to the displacements and damage of the skull and right side of the skeletal remains. To determine reliable scenarios of bead combinations, the archaeological documentation was carefully studied, focusing on preserved alignments of beads (as a "DNA" sequencing approach) to reconstruct eventual "chains". Mathematical analyses were performed to estimate the lengths of these chains. Use-wear analyses allowed the identification of differential intensities of use while providing indications on the hanging system of the beads. The identifications were based on comparisons with key studies describing patterns of use due to bead-to-bead alignments within ornamental compositions e.g. [65–67]. Finally, it was essential to reconstruct the initial shape of the ring, which was very damaged by the burial conditions, as the missing parts may have played important role in the structure of the ornament, for example, the number and lengths of rows of beads. In S3 Appendix, we explain how we reconstructed the shape of the ring, and how we estimated the number and lengths of rows and the overall volume of the ornament.

## Results

### The diversity of raw materials

The materials employed for making the beads are predominantly stones, followed by marine shells. XRF measurements made on samples of stone beads (Fig 5) confirmed the preliminary visual identification of turquoise, hematite, and calcite.

For three calcite beads, the percentage of the chemical elements indicates over 90% of $CaO_3$, which can be interpreted as a highly pure variety of calcite. The reddish nuances are related to the presence of iron (Fig 6).

Additionally, X-Ray diffractometry was applied on the powder of a small fragment from a broken disc bead (Fig 7). It unequivocally also confirmed the calcite. This calcite, which has also been used for tubular and flat beads (Fig 4A and 4B), is soft (<4 on Mohs hardness scale) with compact, relatively homogenous grains that seem to allow the material to transform smoothly without much risk of fracture.

The five green disc beads (Fig 4D) were all identified as turquoise, which is a relatively hard mineral (5–5.5 on Mohs scale). XRF measurements show different percentages of the chemical elements, yet with relatively high amounts of Al, followed by P, which are typical of turquoise (Fig 5). Those found in grave CG7 have a dark, wine-coloured gangue full of translucent crystals, possibly quartz. Turquoise with similar inclusions were recorded from other burials at Ba'ja [59].

Three large items are made from hematite (Fig 4G–4I), a quite dense and hard material (5.5–6.5 on Mohs scale), characterized by its metallic lustre and dark brown to red colour due to its iron oxide composition (Fig 5).

Marine shells found in this assemblage comprise bivalves, gastropods and scaphopod species. Respectively, these correspond to *Pinctada margaritifera*, whose nacreous face (a.k.a. "mother-of-pearl") was used to create the ring (Fig 4F), to small unspecified *Conus* shells and to a unique small tusk shell (Dentaliidae) fragment. For the white banded tubular beads (Fig 4E), it was necessary first to identify whether the material belonged to the stone or shell group. XRF measurements on one specimen indicate an aragonite and calcite composition identical to those of marine shells. To go further and identify the taxa, palaeoproteomics analyses were performed; the intracrystalline shell proteins were extracted from three beads (*Cf*. S2 Appendix for an extended description of the method and data). A secure taxonomic identification was achieved by searching the raw data obtained from high-resolution tandem mass spectrometry

against a large molluscan protein sequence database, assembled from publicly available data-sets. The identified ancient proteins showed characteristics consistent with shell proteins (including domains involved in biomineralization processes such as chitin-binding, vWFA, and disordered regions). Some peptides had diagenesis related modifications, supporting the endogeneity of the sequences [68]. The results mark the oldest mollusc shell proteins recovered and necklaceized so far and show unequivocally that the raw material used to make the ana-lyzed three beads were *Tridacna* sp. shells (Fig 8).

*Tridacna* beads show different alternated matte to translucent bands, which are the growth lines of the shell. The bands have different widths, and also a different organization: they are either tight (very close one to the other) or far apart. This may be either the result of a selection of the material from different locations from the same *Tridacna* valve, or different valves were used to provide these visual effects.

Finally, the ornament includes two unique beads made from an exceptional material, first identified as fossil resin (J. Schultz, State Academy of Fine Arts Stuttgart), and later confirmed as amber (*Cf.* S2 Appendix part II for details). None of the beads' spectra shows the so-called Baltic shoulder—an intense absorption peak in the 1160–1150 cm$^{-1}$ range, preceded by a char-acteristic band between 1250 and 1180 cm$^{-1}$, typical of ambers from the Baltic [69], nor the typical bands of Sicilian amber (simetite) at c. 1241 and 1181 cm$^{-1}$ [70]. Instead, in the C-O stretching region (1300–1000 cm$^{-1}$), a distinctive COOH C-O stretch vibration at 1228 cm$^{-1}$ together an absorption bands at c. 1152 cm$^{-1}$ were recorded. The set of recorded spectral fea-tures in the analysed samples (Fig 9) are compatible with the spectral features recorded for the amber occurring in Lebanon [71, 72].

Never recorded for such an ancient period, these beads represent the oldest discovered archaeologically so far. However, these beads were poorly preserved. Even though the grave filling was systematically sieved, it cannot be excluded that other tiny pieces of amber beads have disappeared or were unobserved during the excavation as the sediments were similar in colour (especially where ochre powder or spots occurred). The analysed samples were used to acknowledge the diversity of mineral raw material used for the ornament. They served as a ref-erential to ID the non-analysed material (Table 1).

## Manufacture and use

Most of the items discovered with the child in CG7 are beads (defined as objects with a centred perforation, Fig 4A–4E, 4G and 4I), that were classified into four typological families (Table 1). Two additional typological groups are represented by single items, the double perforated pen-dant (Fig 4H), and the engraved mother-of-pearl ring (Fig 4F). Table 1 presents the typological diversity in relation to the raw materials. Worth noting is the use of red calcite for three differ-ent types. The hematite is also transformed in two types. On the other hand, tubular and disc beads are made from four and three different materials, respectively.

Before going through each category, it is worth noting that the stone and shell beads share a common general manufacturing process composed of successive shaping, drilling, and finish-ing stages. Based on the study of shell and stone preforms of beads found in other contexts at the site [73], shaping seems to be made by abrasion, although the application of sawing for car-bonate-based materials, or percussion for turquoise or hematite, cannot be excluded. Drilling was systematically made from both sides (bidirectional), regardless of the type and material. However, the finishing stage, which implies the regulation of the shapes and smoothing of the surface, was not systematic and depended on type and material. Some general notes can also be made regarding the use-wear analysis. Micro and microscopic observations indicated a dis-cernible difference in the degree of wear between the turquoise and hematite items, and the

*Measurement Conditions:* (Bookmark 1)

| | |
|---|---|
| Measurement Date / Time | 21/05/2019 15:41:54 |
| Operator | manip |
| Raw Data Origin | XRD measurement (*.XRDML) |
| Scan Axis | Gonio |
| Start Position [°2Th.] | 10,0814 |
| End Position [°2Th.] | 79,9174 |
| Step Size [°2Th.] | 0,0790 |
| Scan Step Time [s] | 141,2700 |
| Scan Type | Continuous |
| PSD Mode | Scanning |
| PSD Length [°2Th.] | 3,35 |
| Offset [°2Th.] | 0,0000 |
| Divergence Slit Type | Fixed |
| Divergence Slit Size [°] | 0,2500 |
| Specimen Length [mm] | 10,00 |
| Anode Material | Cu |
| K-Alpha1 [Å] | 1,54060 |
| K-Alpha2 [Å] | 1,54443 |
| K-Beta [Å] | 1,39225 |
| K-A2 / K-A1 Ratio | 0,50000 |
| Generator Settings | 30 mA, 45 kV |
| Diffractometer Type | 0000000000004271 |
| Diffractometer Number | 0 |
| Goniometer Radius [mm] | 240,00 |

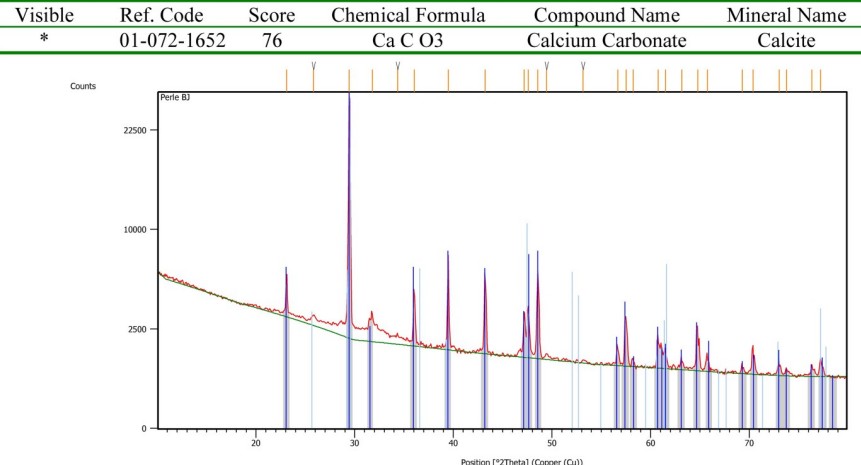

| Visible | Ref. Code | Score | Chemical Formula | Compound Name | Mineral Name |
|---|---|---|---|---|---|
| * | 01-072-1652 | 76 | Ca C O3 | Calcium Carbonate | Calcite |

**Name and formula**

| | |
|---|---|
| Reference code: | 01-072-1652 |
| Mineral name: | Calcite |
| Compound name: | Calcium Carbonate |
| ICSD name: | Calcium Carbonate |
| Empirical formula: | $CCaO_3$ |
| Chemical formula: | $CaCO_3$ |

**Crystallographic parameters**

| | |
|---|---|
| Crystal system: | Rhombohedral |
| Space group: | R-3c |
| Space group number: | 167 |

**Fig 7. X-Ray diffractometry analysis applied on a prepared sample from a disc bead from grave CG7.** Details on the measurement setting and a diffractogram indicating the calcite mineral. Measurement and Graph: G. Monge, CEMEF.

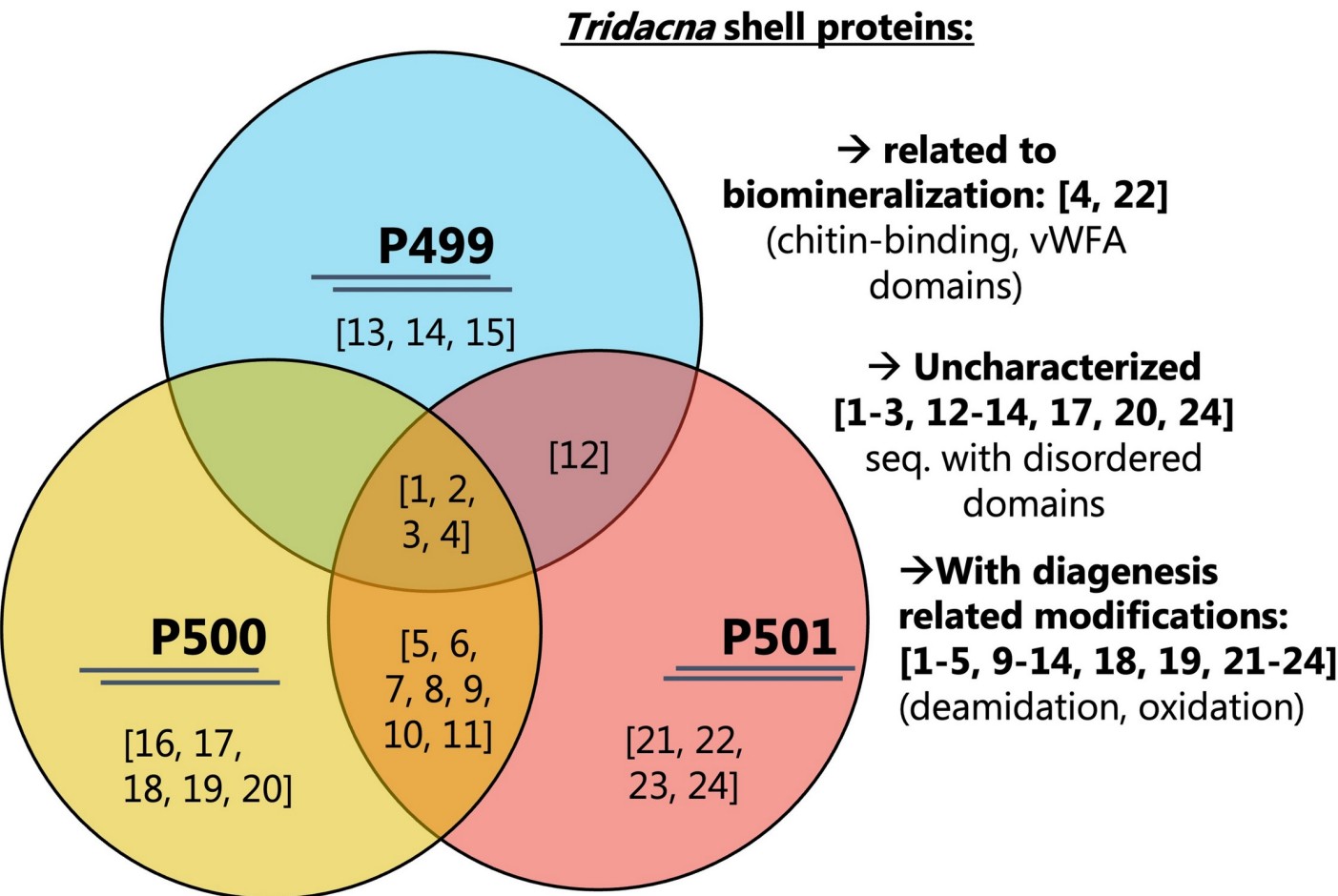

**Fig 8. Schematic representation of the results obtained by palaeoproteomic analysis of intracrystalline protein fraction extracted from archaeological beads.** The Venn diagram indicates the proteins identified in each of the three samples (PALTO499, PALTO500, PALTO501) and those which are shared. The numbers correspond to sequences provided in Table 2. On the right-hand side, we note sequences that are typical of ancient mollusc shell proteins, i.e., with biomineralizing and disordered domains and sequences that were identified with diagenesis related modifications. Author: J. Sakalauskaite.

calcite and shell beads. The first group shows a high intensity of wear in comparison to the calcite and *Tridacna* beads (see below).

## The beads

The disc, tubular, flat, and compact beads occur in different numbers and are made of different materials (Table 1). The relation between types and materials is specific to grave CG7. In other contexts, the same typological families can occur in other materials. In other words, there is no strict norm dictating that specific typological families of beads should be made from specific materials and vice versa. The metric data for the main bead types (GN) are summarized in Table 2.

The diameter and length values of the beads were plotted per type and material in Fig 10. Although the groups are distinguished by the specific morphologies of the beads (e.g., disc *vs*. tube), a general size homogeneity is observed. For over 1530 disc and tubular beads, the diameter ranges between 3 and 6 mm. The variability is more important for the length values of the calcite tubular beads.

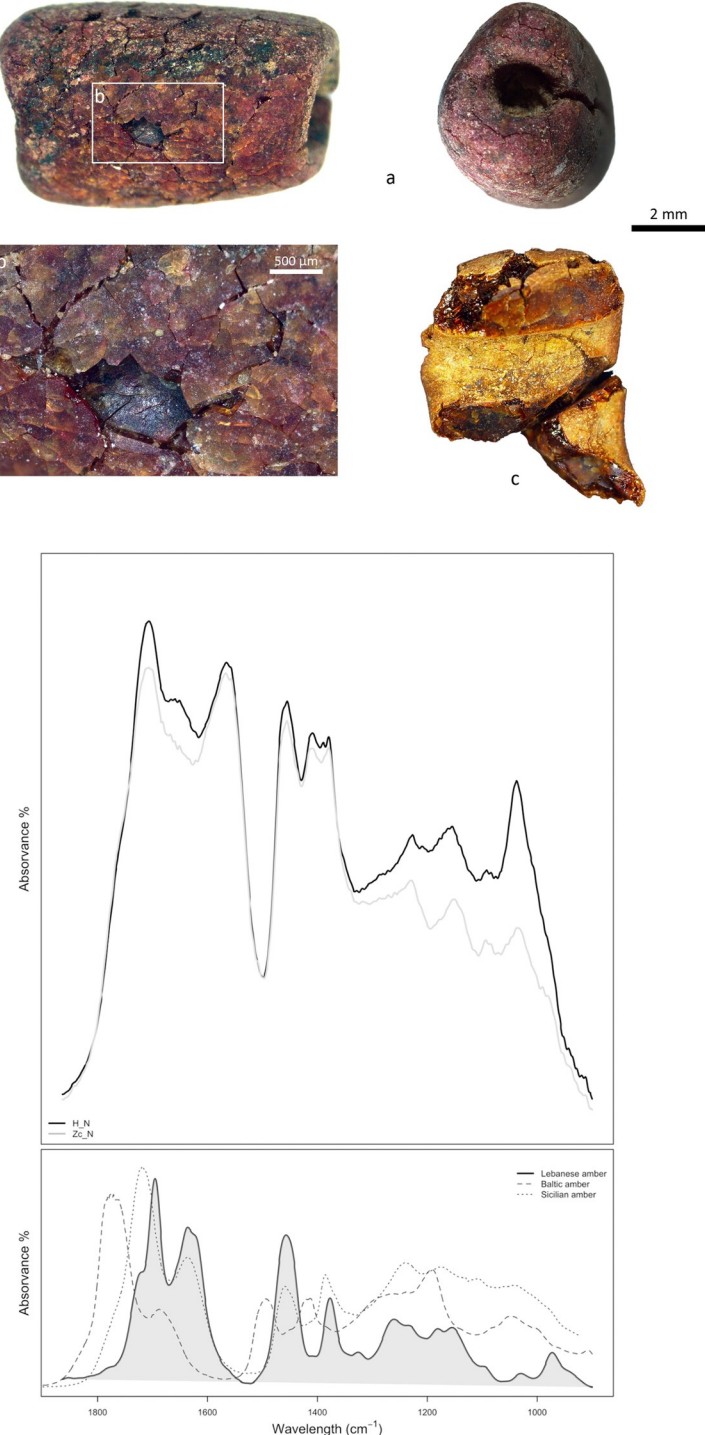

**Fig 9. FTIR spectrum.** FTIR spectrum in the fingerprint region of the analysed samples compared to standard reference spectrum from different origins (Lebanese amber spectra has been kindly provided by D. Azar). Graph: C. Odriozola Lloret, J. Ángel Garrido Cordero, M. Ángel Avilés.

Disc beads are by far the most dominant category with at least 2273 specimens (Fig 10). The most common type are regular small circular discs with sub-rectangular profile, made

**Table 2. Metric values of the main beads' typological families of the child's burial (only for items GN).**

| Beads | Metric data | Length | Diameter | Perforation | Circularity | Roundness |
|---|---|---|---|---|---|---|
| Calcite disc | Nb | 1492 | 1492 | 1492 | 1492 | 1492 |
| | Maximum | 3.67 | 5.67 | 2.302 | 0.908 | 0.999 |
| | Minimum | 0.58 | 3.056 | 0.944 | 0.773 | 0.827 |
| | Average | 1.423 | 4.045 | 1.546 | 0.890 | 0.958 |
| | Standard deviation | 0.347 | 0.391 | 0.223 | 0.012 | 0.026 |
| Calcite tubular | Nb | 46 | 46 | 46 | - | - |
| | Maximum | 10.56 | 4.45 | 3.120 | - | - |
| | Minimum | 4.63 | 3.08 | 1.400 | - | - |
| | Average | 7.26 | 3.802 | 1.982 | - | - |
| | Standard deviation | 1.75 | 0.299 | 0.410 | - | - |
| Shell tubular | Nb | 126 | 126 | 126 | - | - |
| | Maximum | 14.24 | 5.86 | 3.060 | - | - |
| | Minimum | 4.49 | 2.8 | 1.240 | - | - |
| | Average | 9.34 | 4.681 | 2.070 | - | - |
| | Standard deviation | 2.27 | 0.698 | 0.339 | - | - |

For all, the standard deviations (SD) of the diameter and perforation have low values, indicating a standardized diameter and diameter of perforation per type. In the case of disc beads, even the length (= thickness) values are highly standardised (low SD), meaning that their manufacture has required significant control.

from reddish calcite (Fig 11A–11C). Beads with irregular sub-oval to polygonal shapes, made from turquoise (Fig 11F–11I), were also considered as discs, along with those identified as circular slices of *Conus* shell (Fig 11D and 11E). The diversity of materials used offers a play of contrasting colors between the dominant red and the white.

Calcite disc beads were manufactured according to the general three stages scheme mentioned above. Regarding the shaping stage, we have no evidence for the way the disc preforms were obtained, whether through sectioning long tubular drilled beads into segments (discs), or via shaping by abrasion preforms of discs, one by one, and then drilling and finishing them. The latter procedure is well documented at PPNA (Bangsborg-Thuesen et al., submitted) and PPNB [74] sites in Northern Jordan where stone bead workshops are attested. However, the preforms of stone disc beads remain unrecorded at Ba'ja. Furthermore, no diagnostic patterns of sectioned discs from long beads [75] were observed for the disc beads from burial CG7. Most of their perforations have a bi-conical section. The cylindrical section is recurrent. We interpret this section as an intentional enlargement of the initial hole, probably made during the threading process of the beads. Intensive use-wear as the cause of this section is excluded as it would be accompanied by a strong rounding of the edges of the perforations, which is not the case. In addition to the standardized size of disc beads (Fig 10), the circularity and roundness degrees are very close to 1 (Table 2), meaning that most of the beads have perfect circular outlines and rounded shapes.

When plotted with the diameter, both the circularity and roundness show higher values for smaller beads than for the large ones, as indicated by the trend curves (Fig 12). This is understandable as small diameters could reflect a more intensive finishing process. Compared with the circularity, the roundness values are higher (closer to 1 = more regular). This is also understandable since this value refers to the general shape, which is less directly affected by the final polishing/abrasion of the beads' outlines. Low values (closer to 0) of circularity and roundness correspond to beads presenting fractures on the outlines or particular shapes (Fig 11C, last 3

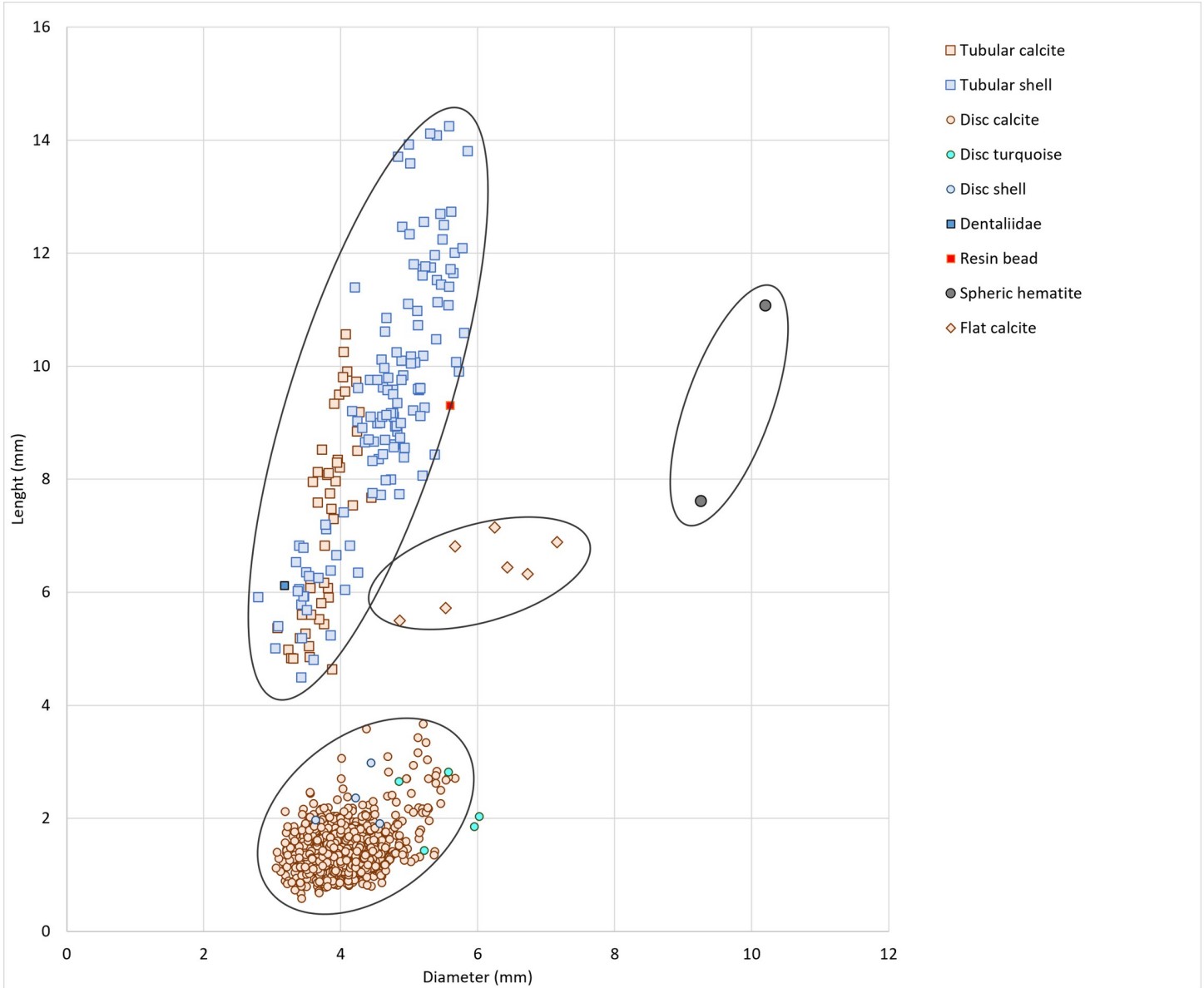

**Fig 10. Proportions of length to diameter of beads (GN) according to the typological families and materials.** Graph: H. Alarashi, Ba'ja N.P.

beads to the right), which might have been produced due to prolongated process of finishing on the same spot.

The standardized size and regularity of the shape and outline of the calcite disc beads are consistent features with batch processing, *en-masse* [74, 76] that consists in polishing the series profile of beads strung together. However, experimental work is needed to verify the efficiency of batch grinding using different tools and auxiliar agents (water, abrasive, etc.), and to test which technique provides more qualitative results and regular shapes (batch processing of strung beads or polishing singular beads manually?).

The irregular shape of the turquoise beads (Figs 4D, 7 and 11F–11I) strongly contrasts with those of the calcite disc beads. This is however not specific to this assemblage as all the turquoise beads found at the site, in and outside funerary contexts, have irregular shapes [58, 77].

Could the quality of the material or the presence of (quartz-like?) inclusions (Fig 11J–11K) have caused technical constraints that made the final stage of the transformation more challenging? While this might be the case, the technological abilities of the bead-makers are not questioned. Instead, it is the degree of their investment in the finishing stage, which is closely linked to the quality, that should be considered.

Unlike the calcite disc beads, the turquoise beads have a shiny and smooth surface, with highly rounded edges and complete disappearance of the manufacturing marks, which is diagnostic of intensive wear. The inner surfaces of the perforations also show intense and homogenous smoothing and polish, suggesting a free suspension of the beads, *i.e.*, they were not fixed through knots or stitches.

The *Conus* discs (Fig 11D and 11E) are cut slices corresponding to the largest diameter of the shell whorl and having a missing (removed intentionally?) apex allowing the perforation. The poor preservation did not allow the identification of any manufacture or use-wear traces properly. The study of well-preserved *Conus* discs found at Ba'ja in other contexts indicates however that the apex is eliminated by abrasion and the whorl cut by sawing [73].

With a total of 298 specimens, tubular beads are the second important typological family of the assemblage. Two types were roughly distinguished between cylindrical (Figs 13F and 13G and 16E) and "barrel"-shaped (Fig 13A–13E). For many beads the difference is very subtle and depends on the side of the profile we observe. The section is sub-circular to oval. Most of the tubular beads are made from valves of *Tridacna* shells (Table 2). The other material is the same reddish calcite (Fig 4B) employed for the disc beads.

*Tridacna* fragments were transformed into tubular beads considering the natural alternated opaque/ translucent growth lines, resulting in three general patterns, with horizontal (Fig 13D–13G) being the most common, then oblique (Fig 13C) and longitudinal (Fig 13A and 13B). The length values of these beads are more variable than those of the calcite tubular beads (Fig 10), with a mean value of 8.5 mm and only a few items longer than 10.5 mm (Table 2). The relation between the length and the patterns of alternating bands was examined. A one-way ANOVA attests that there is a statistically significant difference in length (F = 5.381; p [same] = 0.00583) between groups of beads displaying different patterns of bands (growth lines). Results of the Tukey post hoc test (F = 4.576; p[same] = 0.004475) indicate that the beads with longitudinal pattern are significantly shorter (mean L = 7.30 mm) than the beads with horizontal pattern (mean L = 9.72 mm). No significant difference was observed between these latter and the beads with oblique pattern (mean L = 9.13 mm). One interpretation of this relation could be that when the Tridacna fragment was not long enough to provide beads with a horizontal (or oblique) pattern, the material was exploited differently, that is, transformed into beads showing a longitudinal pattern. When the *Tridacna* beads were arranged within the ornament, the differences in the patterns of their bands might have increased the dynamic and attractiveness of the combination.

Longitudinal parallel striations (Fig 14A and 14B) are observed on the surfaces of the beads, indicating their polishing, which generally corresponds to the final stage of the manufacture. Parallel deep striations are also registered on the walls of perforations (Fig 14C). In several cases, these striations are interrupted by the circular concentric striations of the drilling stage, indicating that the perforation took place after the shaping stage of the bead. The outlines of the perforation rims are regular and circular (Fig 14D and 14E), and the perforation walls are marked with parallel concentric regular striations (Fig 14C–14E) typical of fast-drilling techniques such as a bow-drill or a pomp drill.

Perforations were made by bidirectional drillings that cross the length of the bead at relatively even distances (Fig 15). Images obtained from micro-CT-scans of two beads, one shorter than the other, show sub-conical sections of the drilling tubes (Fig 15 centre images up and

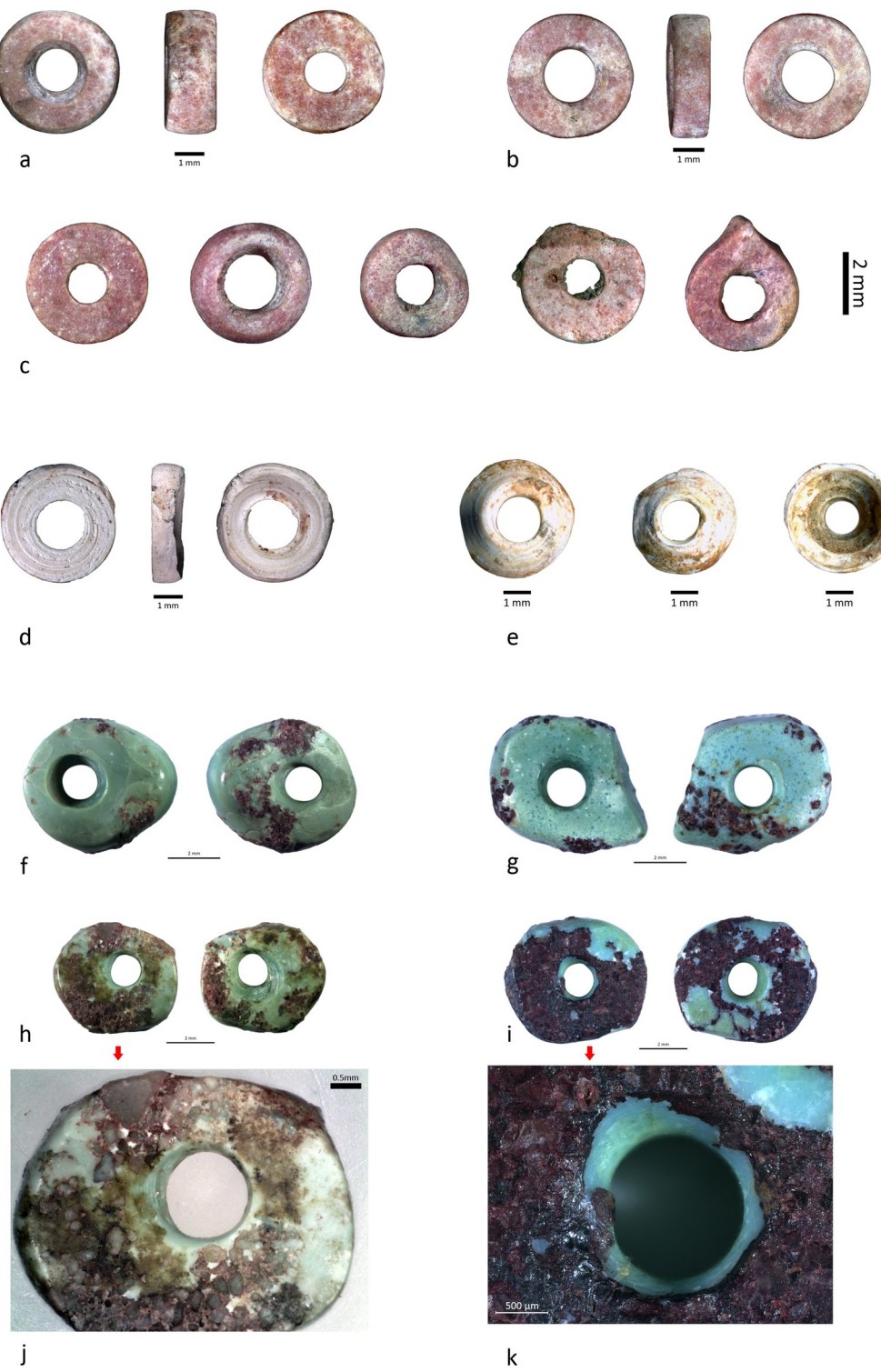

**Fig 11. Diversity of the disc beads and examples of technical and use-wear marks detected on the turquoise beads.**
A-C) Calcite circular disc beads, note that the last three examples to the right show variation in the degree of circularity due to the finishing process by batch abrasion. D-E) *Conus* disc beads. F-K) Turquoise beads. J) Zoom of picture H left. K) Zoom of picture I left. Photos: H. Alarashi, Ba'ja N.P.

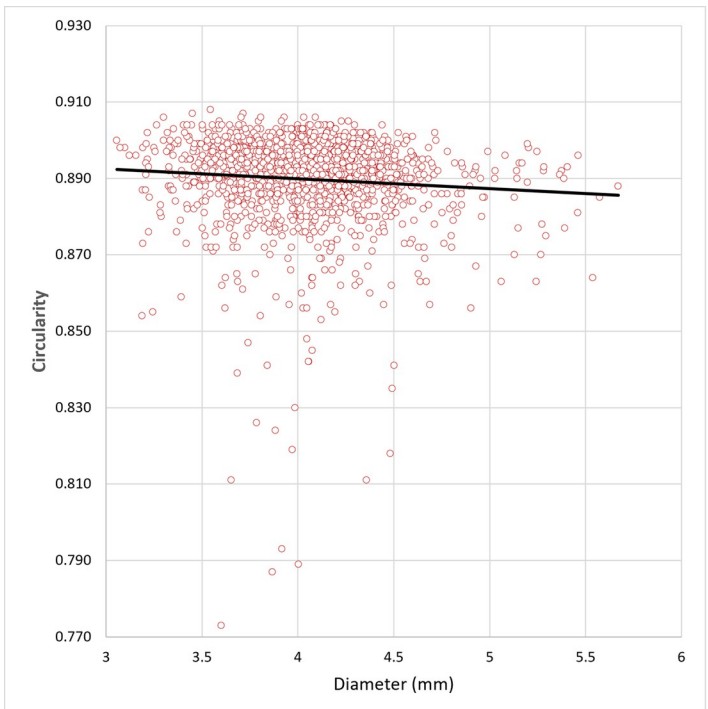
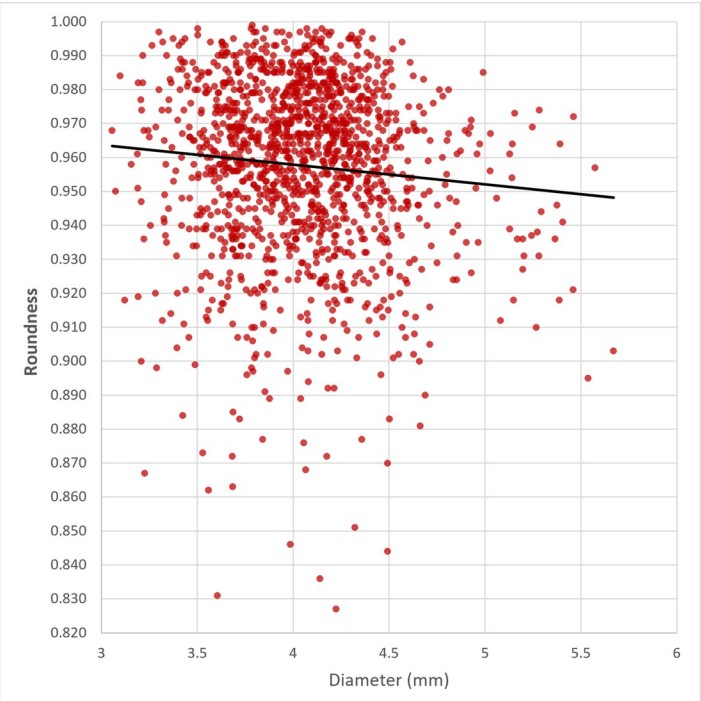

**Fig 12. Relation between the diameter of the calcite disc beads and the degrees of their circularity (left) and roundness (right).** The closer the degree to 1, the more regular is the outline (circle) and the shape (disc) of the beads.

below), which in fact reflect the shape of the drilling bit used. The alignment of the bidirectional (tubes) drillings is almost perfect, or slightly overlapping. The openings of the drillings (Fig 15, right images up and below) show a rather regular and well oriented trajectory. Images made in a transversal section (Fig 15, left up and below) show the constant regularity of the circular shape of the tubes. These features confirm the application of fast drilling technique of soft materials such as shells or carbonate-based stones that we have reproduced experimentally (Pichon, Alarashi 2018).

The alignment of drillings is indicative of experience and skills of the bead-makers. Indeed, turning the bead and drilling it from the opposite side to meet the end of the first drilling perfectly is a complex procedure [78, 79]. Moreover, when considered that the beads were first shaped into cylindrical/barrel-shaped volumes and then drilled, as suggested earlier, it might indicate that the artisans were quite confident in their ability to perform the drilling at the end. The same behaviour is documented at the Neolithic site of Çayönü Tepesi in the Tigris valley, where preforms of cylindrical and flat beads were discovered perfectly polished but not yet drilled (Alarashi, personal observation).

While most of the beads show traces of ochre on their surface (Fig 13B–13G), others have also significant amounts inside the perforation (Fig 14C–14E). Ochre residues were also detected on *Tridacna* beads found in non-funerary contexts [73]. Whether this deposit is due to the use of strings covered with red ochre to thread the beads, to the presence of ochre on the body or clothes of the child, or to other actions related to the funerary event is still an opened question. It is worth mentioning that a lump of red ochre was deposited near the legs of the child in grave CG7. More generally, the use of ochre is attested at the site as powder or liquid spread over collective burials, bones, and artefacts, but also through coloring of human bones, and in one case (CR17:117), through a complete coloring of a cowry shell [58, 77]. It is

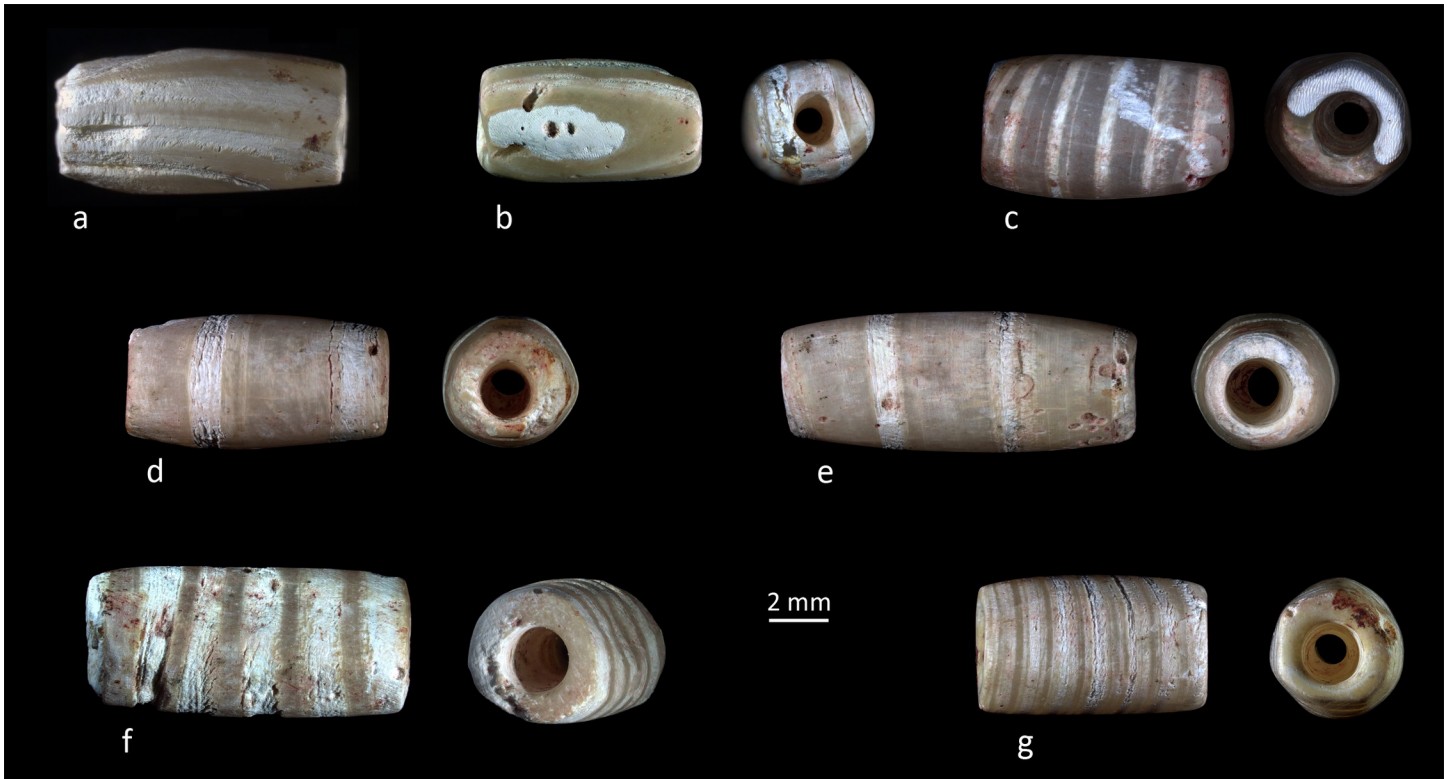

**Fig 13. Examples of *Tridacna* shell beads from grave CG7 showing three patterns of growth lines.** A-B) Longitudinal. C) Oblique. D-G) Horizontal. Photos: H. Alarashi, Ba'ja N.P.

therefore unclear whether the use of ochre on the beads was intentional or as a consequence of the burial rituals. Studies of beads from other burials will possibly help understanding the role played by the ochre into the ornamental practices at Ba'ja.

Finally, *Tridacna* beads show low intensity of use as indicated by a shallow rounding of the rims of perforations and the intersectional edge between the face of the perforation with the profile of the bead (Figs 13G and 14C–14E). Scratches and/or polish are generally more frequent on the perforation faces than on the bead profile. This is likely due to the friction caused by the bead-to-bead alignment in a chain, and to the friction with the string between them. If these beads were used during a lifetime, the wear was not intense enough to completely erase the technical traces, especially the drilling striations inside the perforations, with their circular continuity being still visible.

Calcite tubular beads (Fig 16D and 16E) were manufactured according to the same method as the calcite flat beads. They have use-wear marks on the extremities, evidenced by a partial obliteration of the drilling striations with a polish, or with polish and thinning of the perforation edges (Fig 15F). These marks are known to be produced by a bead-to-bead contact and friction, which is consistent with the chain-like general arrangement of the beads that sometimes results in the extremities of elements into those of their neighbors.

The seven beads classified as flat were made of red calcite and have similar sizes (Fig 10). The profile is sub-oval with a lenticular cross-section (Fig 16A and 16C). So far, only one similar calcite flat bead was found at Ba'ja, in the multiple burial CG9 containing four subadults (F. no. 110825.850). Flat beads with lenticular cross-section made of *Tridacna* shells were also discovered in both funerary and other contexts at Ba'ja [73, 80]. Use-wear marks are very similar

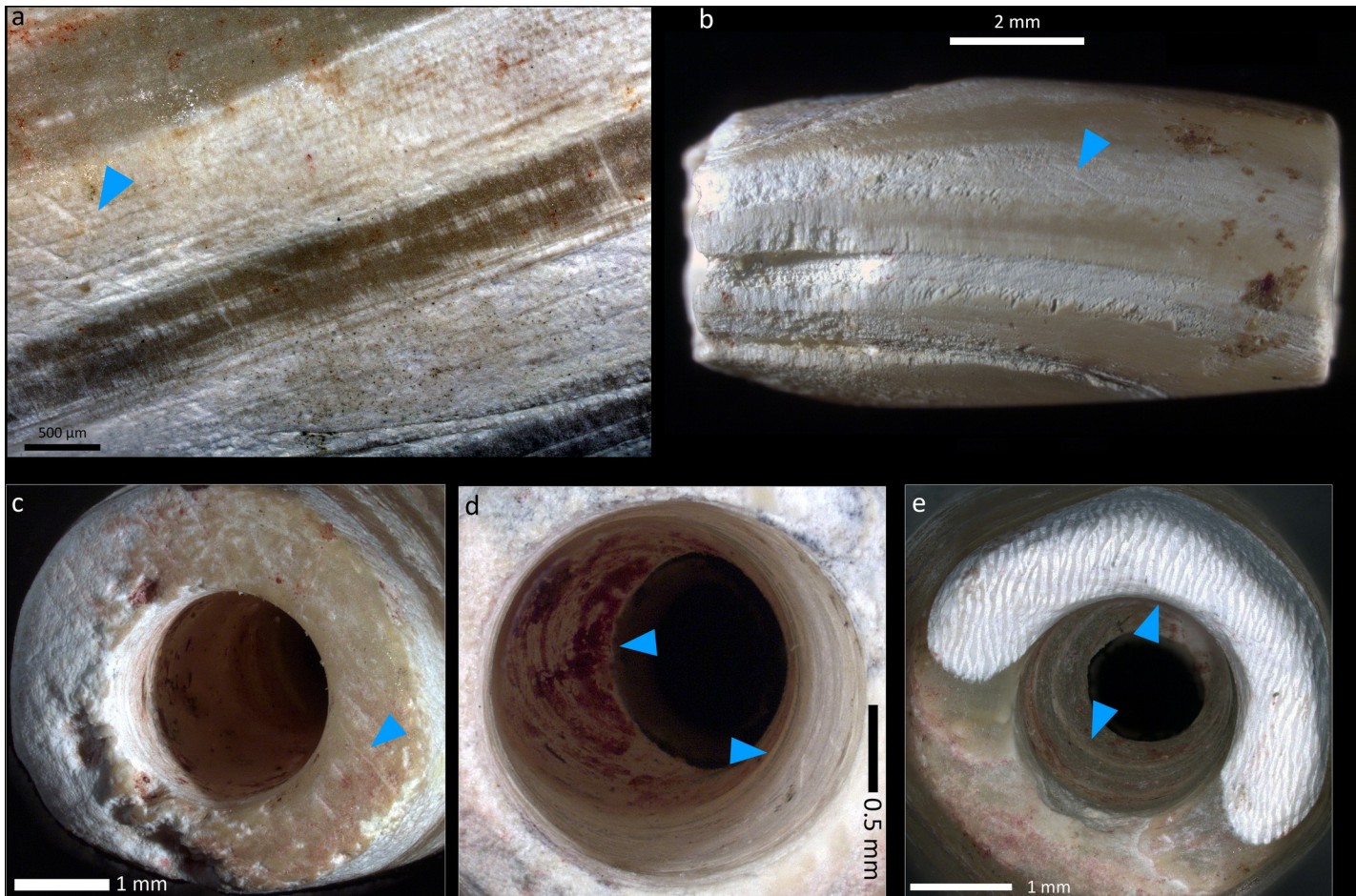

**Fig 14. Technical and use-wear traces observed on the *Tridacna* shell beads.** A-B) Deep oblique and longitudinal striations of abrasion. C) Parallel striations of abrasion on the perforation face. D) Concentric and regular drilling striations, right arrow. See also the remains of red ochre covering the "walls" of the drilling and inside the concentric striations, left arrow). E) Circular and regular rim of the drilling at the opening (surface of perforation, upper arrow), and at the end (junction, meeting area). See also the concentric regular deep drilling striations (lower arrow). Superficial use-wear is observed by weak and little pronounced rounding of the rims of perforations in C-E. Photos: H. Alarashi, Ba'ja N.P.

to those observed on the extremities of the tubular beads around the perforation edges. These areas show a slightly different hue of red, possibly due to the use wear. Such use-wear color modification is well-known for shell, bone and tooth ornaments [66, 67]. It can also occur on different kind of stones, namely soft and hard varieties.

The typological family of compact beads is represented by two spherical hematite beads (Fig 17) similar in size (Fig 10). The transversal section is sub-circular, and the color is brownish grey to dark red. This type has one comparable bead from the LPPNB site of Basta, from Loc. B68:13 (F.no. 10835; Hermansen n.d.). More frequent parallels can however be found in the Northern Levant, notably in the Middle Euphrates. One example is quite interesting as it is very similar in shape, size, and color, yet it comes from a PPNA context from Tell Mureybet [81] fig 18.13. Spherical beads made from carnelian and agate were also found in a burial of a child at Tell Halula [64].

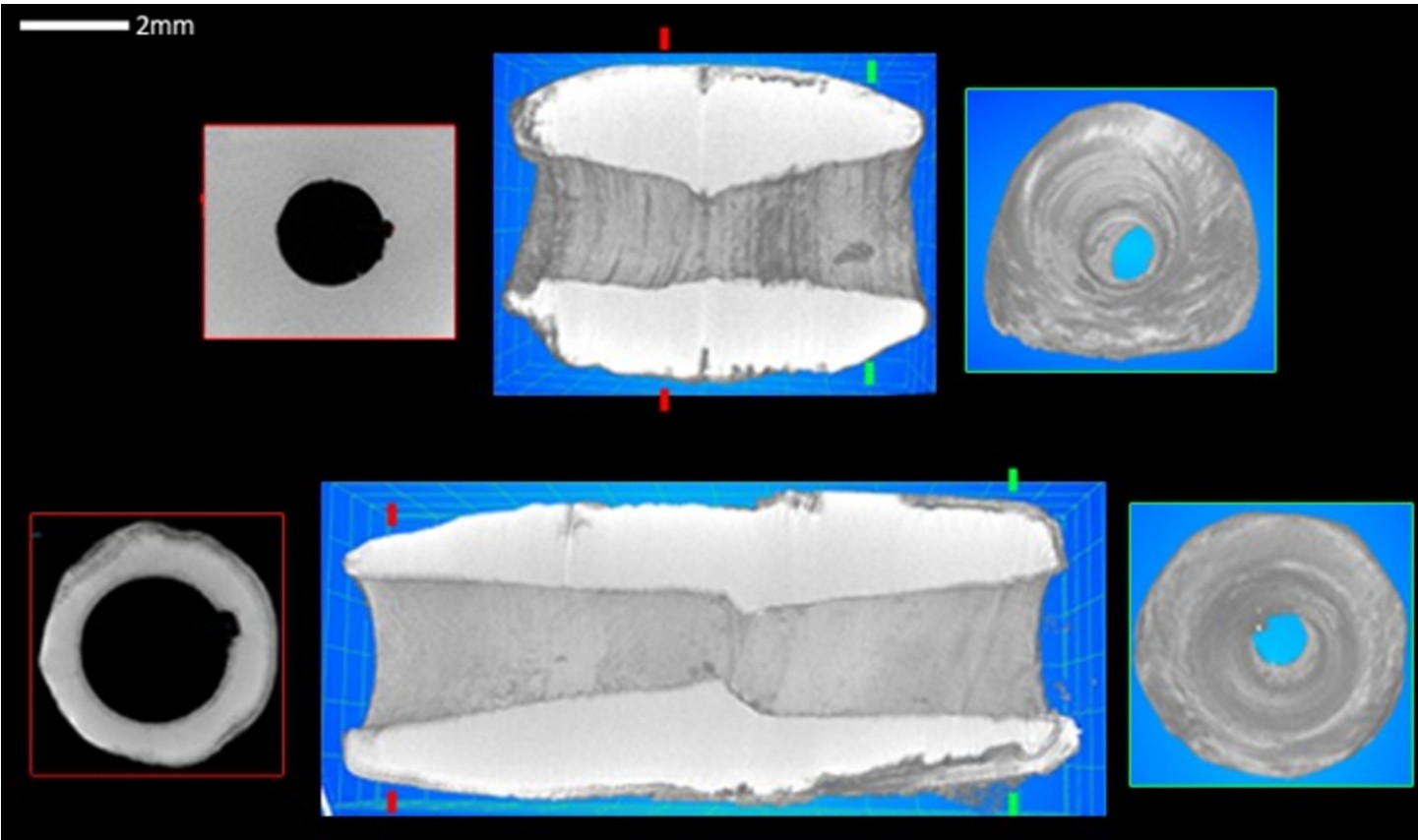

**Fig 15. Images obtained from micro-CT scan of two *Tridacna* beads.** The green marks on the central up and below images indicate the transversal sections of the images to the right, while the red marks indicate those to the left. Composition: H. Alarashi, picture acquisition: L. Vigorelli and A. Re.

## The double perforated pendant

This exceptional object was found behind the neck of the child with many disc beads above and below it, and with some beads still sticking to the perforations. Because of its position, it was originally designated as a "buckle", a function that was subsequently confirmed by use-wear analysis (*see* below). This flat, oval-shaped plaque is made of dark-grey hematite and has lateral double perforation (Fig 17A and 17B). While other double-perforated stone objects were found at Ba'ja, they are primarily made of turquoise [77]. Moreover, none of these artifacts are comparable in terms of the regularity of the shape, size, and quality of finishing, especially given the hardness of the stone used for this plaque.

Multidirectional deep striations are observed all over the surfaces of the buckle and of the compact spherical hematite beads. The latter also show impact points and scratches everywhere (Fig 17I and 17J). Despite the density and hardness of the material, the buckle was intensively worn. The surface shows shiny polish (Fig 17A, 17B and 17H), very intensive rounding of the edges, and enlargements of the perforations towards the external edges (Fig 17C–17F) in two opposite directions, as if the perforations were under significant tension. These observations support the hypothesis that this object was used as a "spacer" or "buckle" to hang strings of beads behind the neck. They also argument for a long-time use within an ornamental composition that have provoked similar wear makes.

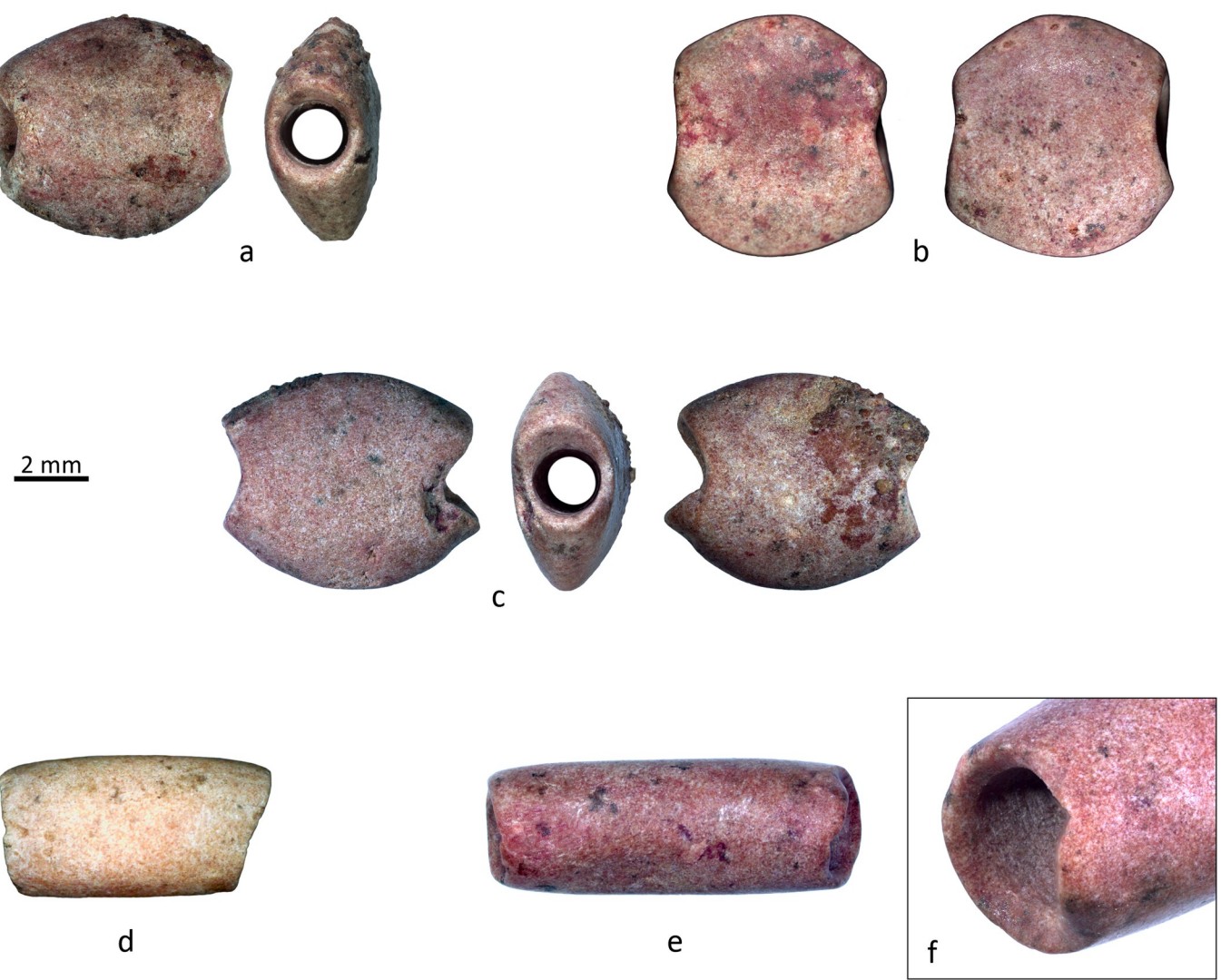

**Fig 16. Examples of flat and tubular calcite beads from CG7.** A-C) Examples of bi-truncated flat beads with lenticular section. D-F) Examples of tubular calcite beads of a cylindrical type. F) Use-wear observed through erosion of the drilling striation, thinning and fracture of the walls of the drillings due to friction and insertion of the extremity of the neighboring bead into the perforation. Photos: H. Alarashi, Ba'ja N.P.

## The decorated ring

The mother-of-pearl ring is poorly preserved and presents fractures mostly on the decorated parts. Nomenclature and measurement convention are given in Fig 18 and the metric values in Table 3.

The ring was found in a vertical position in the grave. Fig 18 presents it in the same orientation as it was found. The hypothetical reconstruction (details are provided in S3 Appendix) of its shape is based on morphometrical analysis of the preserved parts, and on comparisons made with similar items discovered years ago at Ba'ja (Fig 19) and Basta ([82] figs. 10, 12; [57] fig. 13.3, Hermansen n.d.), although the example from this latter site is much smaller. It is worth noting however that the ring from Basta was likewise discovered with many beads very similar to those of Ba'ja, all associated with a child (*cf*. up-cited references).

The microscopic observation of the ring revealed series of marks related to the manufacturing process. A technological study of the mother-of-pearl objects along with unworked *Pinctada* valves, slabs, debris of manufacture and other fragments found in non-funerary contexts at Ba'ja allowed the main manufacturing stages performed to produce mother-of-pearl rings to be traced [73]. On the convex part of the ring, parallel abrasion striations (Fig 21C) indicate the removal of the periostracum layer in order to reveal the nacreous surface. This corresponds to the earliest stage in the preparation of the shell nacreous slab from which the ring will be cut. On the outer and inner circumferences (Fig 18: C.1) of the ring, series of curved striations (Fig 21D) indicate the extraction of the ring from the slab. Finally, the sawing/engraving marks observed on the outer circumference (C.3) indicate how the denticulated shape of the outgrowths (Fig 21E) was obtained. The perforations on the outgrowths were made from both sides by drilling, as attested by concentric striations still visible (Fig 21A). Some of the perforations show very regular circular holes and concentric striations that might have been the result of fast drilling using a bow-drill. Here again, experiments are needed to infer the technique applied. In all cases, drilling several holes on the delicate, thin denticulated outgrowths of the ring undoubtedly required a very careful motion, high skills, and adequate tools.

The ring manufacture produced a non-negligible number of "wastes". The extraction of the inner circumference (C.2) would have provided a relatively large piece ($\sim$ 30 mm diameter), with which it is possible to create a range of other objects: buttons, small rings of the simple flat type, etc. e.g. [73, 77]. On the other hand, the C.3 of the ring in which the outgrowths are ascribed (>60 mm) indicates that the initial shell slab should have been quite large to allow the extraction of a relatively flat ring.

Use-wear analyses were inconclusive because of the poor preservation of the ring. The upper layers of the mother-of-pearl have disappeared almost completely in some areas. It was therefore not possible to determine whether the deformation (enlargement) on the rims of some perforations (Fig 21B), or depressions between the perforations (Fig 21F) were due to the frictions and tension exercised by the string on these areas or not. Use-wear analyses made on other rings and mother-of-pearl items from the site [73] show, however, that when multiple perforations are present, these show clear use-wear (enlargement, rounding, erosion of layers, etc.), meaning that these perforations were functional and not merely decorative. In the case of the child's ring, beads were still stuck to some perforations (Fig 21G and 21H). A possibly organic material with ochre residual is still imprisoned inside the perforation of the ring and the bead. It might have been the remaining part of the string.

## Reconstruction of the necklace

During the excavation (see details in S1 Appendix), we documented the concentration of beads around the cervical vertebra, in front and above the ribs, around the manubrium bone, beneath and around the mandible and next to the occipital bone, with few scattered specimens around the facial bones of the skull. This distribution indicated that the beads were meant to adorn the upper part of the child's chest and the neck, that is, they were not part of other ornament types such as headdress, coiffe or diadem. Unfortunately, the inhumation choice of laying the body on its left side did not preserve all the items in their original positions. As a result, a significant portion of the ornamentation, particularly on the right side, had collapsed and become mixed with other beads over time. Thus, our challenge was to reconstruct the original combination of the ornament using the most reliable scenario possible. Relying on the analyses of the excavation documentation, the biographical characterization of the objects and on series of estimations and logical considerations, we were able to propose a reimagination of the structure and how the ornament would have looked when displayed on the body.

The orientation of the ring, the number and length of the rows, and the arrangement of beads per row were critical in establishing the overall structure (Fig 22) which we have identified as a necklace.

## The role of the mother-of-pearl ring

The mother-of-pearl ring's large size, delicate engraved shape, and shiny surface, as well as its central position on the chest, indicate that it was the "masterpiece" of the necklace. The equidistant perforations on the outgrowths were not only decorative but also functional, as evidenced by the disc beads that are still connected to them. The ring was therefore the structuring element from which rows of strung beads have spread to adorn the chest and neck of the child. It may have divided the necklace into two even left and right sides. However, its orientation on the chest is crucial for understanding not only how the ornament was displayed, but also for determining the number of rows and their respective lengths. The engraved ring with its four outgrowths can be oriented in a Greek (+) or a Saint Andrew's cross (X). Several arguments support the first possibility. First, the presence of two additional perforations on outgrowth I (Fig 18) suggests its upright position, which corresponds to the position of the ring when it was discovered (Fig 3C) in relation to the body laid down on the left side. Secondly, the engraved ring that was discovered in 2001 (Fig 19) is identical in shape and also has two additional perforations on one of the outgrowths. This suggests that this type bears this characteristic, which is a clear hint to the orientation of these objects. Thirdly, several large mother-of-pearl items discovered at the site have explicitly the shape of a Greek cross and bear equidistant perforations and engraved edges ([6] figs. 4 and 5). Fourthly, if the ring was displayed in an X orientation, it would have implied that the rows spreading from it were not only over the shoulders but also beneath the arms, extending to the back of the child. However, no beads were found on the lateral sides of the or torso below the arms, meaning that the X scenario was not the one employed.

## Number for the rows

The estimation of the number of the rows was initially based on the number of perforations of the ring (18), considering that the beaded strings were attached to these perforations. The very high number of beads, plentiful enough to fill many rows (see. Below), comforted this hypothesis. However, a nine-rows scenario (18 strings, 9 to the left and 9 to the right) would require a constrained display. The placement of the perforations on the ring was a key factor in determining the lengths of the strings, which were all gathered behind the neck. That is, the length of strings was graduated. Those attached to outgrowth I of the ring, the upper one, are shorter than those attached of the lateral outgrowths (II and III). Those attached to the fourth outgrowth in the bottom were consequently the longest. However, using the fourth outgrowth to attach strings that extend all the way to the neck would cause the ring to tilt or incline forward unless it was affixed to a support (cloth?). Moreover, this scenario implies moving these strings aside, beyond those attached to the lateral outgrowths, and fixing them in place. Otherwise, they will hamper the view of the masterpiece, the ring, and the other beaded strings. Thus, a scenario with 18 strings (9 rows) is not optimal from an equilibrium viewpoint. It is also not supported by the distribution of beads in the grave, which were much more concentrated in the upper front of the chest than towards the arms. This brings us to propose another scenario based on 14 strings (7 rows) attached to the ring and spreading towards the neck. This configuration uses the 14 among 18 perforations, only those of the upper and lateral outgrowths (Fig 22). Rows that spread from these perforations can be gathered comfortably behind the neck through a "free" display, without any form of fixations to a support or special arrangement. It

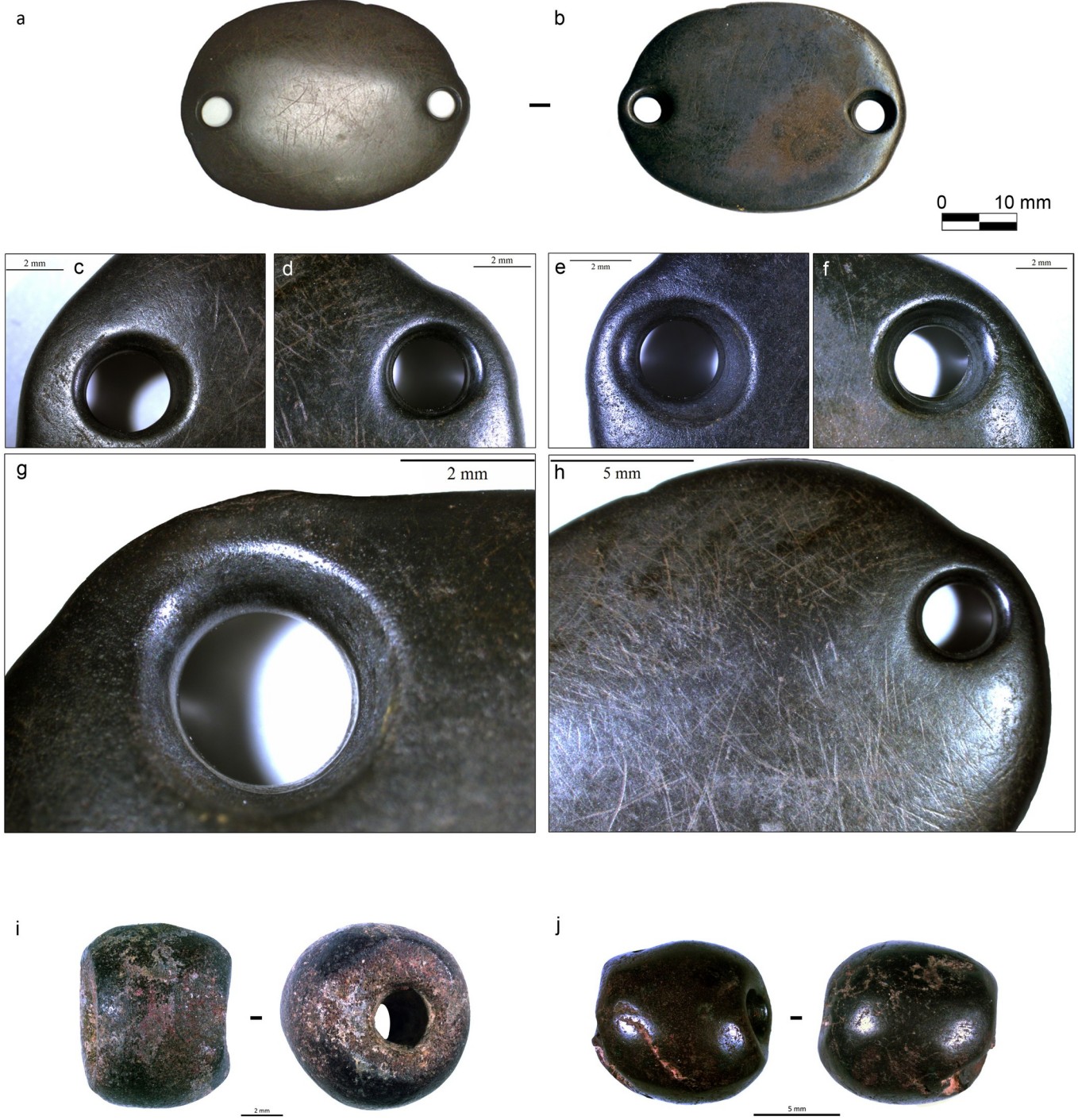

**Fig 17. Hematite objects from grave CG7 with indications of manufacturing and use-wear traces.** A-H) Double perforated pendant. I-J) Spherical beads. Photos: H. Alarashi, Ba'ja N.P.

also ensures an optimal visualisation of the ring and the other rows. As for the fourth outgrowth, it might have been used to hang vertically short strings of beads, or other unpreserved items. Although it is impossible to confirm one or other configuration, a necklace composed with 14 strings attached to the ring appears the most balanced, and more practical as it does

not require further adjustments for an already complex structure. Thus, the reconstruction adapted here is based on this configuration.

The shape and number of perforations of the ring were however not the only criteria that have determined this structure. Three other rows, not connected to the ring, were added (see below) based on other estimations.

## Estimation of the total length

A row connected to the ring is composed from two strings, one on each side. The position and length of each row was determined by the ring perforation to which it was fixed, meaning that the lengths of rows gradually increased going from the central outgrowth towards the lateral ones. Before the estimation of the length of each row, it was necessary to calculate the total length of the necklace when all the beads are tightly aligned one after the other, in a chain-like arrangement. The result of these estimations is impressive as it gave a total length of around six meters, with few centimetres of error rang (Table 4).

For details on the method and formulas used for the estimation of the non-measured beads (GS) and the total length, *cf*. S3 Appendix).

## Estimation of the rows' lengths and readjustment of their number

The calculations were made to estimate the required length of the rows to accommodate all the beads and reconstruct an ornament that would fit the body of an eight-year-old child. Thus, measurements of the upper bodies of children of both sexes between the ages of seven and nine were considered using standard size charts as reference (*cf*. Table 4 in S3 Appendix). Using measurements of the neck, shoulder breadth, and the distance from the 7th cervical vertebra to the waist (both in the front and back), it was determined that the maximum width of the necklace (the distance between the outermost rows) and its length (from the buckle to the fourth extension of the ring) should not exceed 30cm each. Osteological analyses indicate that the child had a normal size and that a necklace measuring 30*30 cm would have fit comfortably the upper body.

When the total length of the necklace was divided to constitute 14 strings of gradual lengths (see above), the estimations gave unsuitable results, as 14 strings were found to be insufficient to account for all the beads. In fact, the total length of 14 strings was less than 4.5 m, which only allow for the integration of 2138 beads. To integrate the remaining 446 beads, over 1.5m had to be added. Therefore, the integration of additional three rows graduating in lengths was considered (Fig 22). The configuration with 14 strings (7 rows) resulted in a significant gap in the areas corresponding to the front of the neck and upper chest, which happened to be the regions where most of the beads were concentrated, as documented during the excavation. However, it was unclear whether these rows were connected or not to the ring, in this case to the upper outgrowth I. The presence of additional perforations in this outgrowth is an argument in favour of the connection of the upper rows (R1, R2, R3) to the ring although the possibility of their independent from the ring cannot be excluded. Especially when considering that a large amount of *Tridacna* beads, especially the big ones, were precisely concentrated in this area of the upper part of the chest, a privileged zone to best exhibit such remarkable beads that show variable patterns of banded decoration (Fig 4E). In other words, connecting the upper rows to the ring would instead highlight the ring and reduce the dynamic effects provided by the *Tridacna* beads. A good compromise that allows to highlight the attractiveness of these beads would be the creation of some discontinuity, namely, disconnecting the upper rows from the ring as proposed in the general structure (Fig 22).

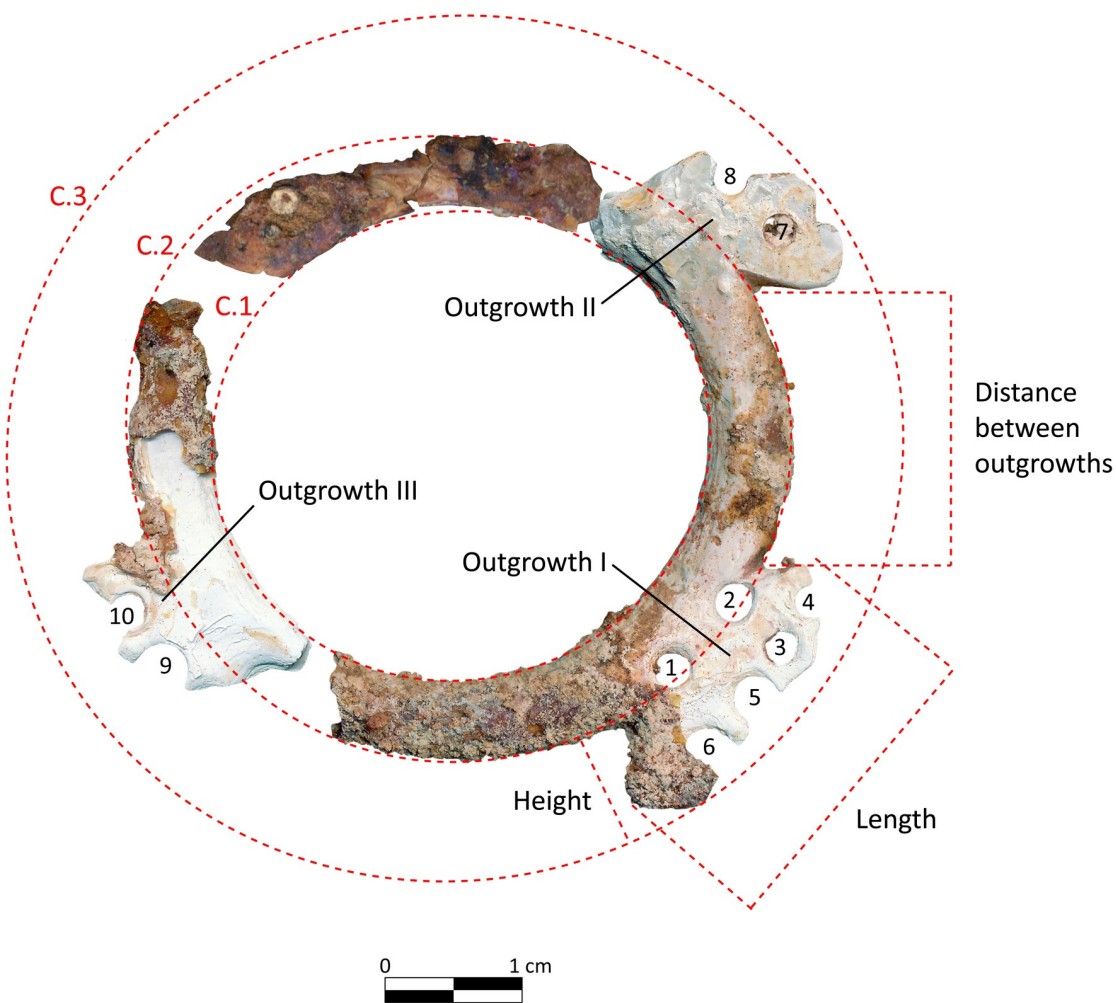

**Fig 18. Nomenclature and convention of measurements used for the study.** C.1 = inner diameter of the ring, C.2 = middle diameter, C.3 = outer diameter. Each perforation has a number. The engraved parts with perforations are called here "outgrowth" (abbreviated "Outg."). Disc beads were found still stuck to perforations N° 7, 4, 9 and 10.

Whether the upper rows were connected or not, the most plausible number of rows turned out to be finally ten, seven connected (14 strings) and 3 separated from the ring. The lengths of rows with details on the number/lengths of beads are given in Table 5 for the main three types: the circular calcite disc, the tubular shell and the tubular calcite beads.

These estimations and the hypothetical structure with 10 rows, opting for the non-connected central ones, were used to physically reconstruct the necklace that was returned reconstructed to Jordan and now exhibited in the new museum of Petra.

## Placing the rare beads

In addition to the main bead groups (Table 5), a total of 22 beads from different types and material (average of total length when aligned = 12.70 cm) were integrated in the combination, respecting their position when discovered as well as functional, typological, or material criteria. The five turquoise beads were distributed within five rows, in the same order of appearance during the excavation, from the top (right) to the bottom (left side). The distribution of these few, yet exotic turquoise beads in different places may have intended to provide dynamic visual

**Table 3. Measurements of the ring.**

| Measurements | mm mm |
|---|---|
| Outg. I P1 | 2.7 |
| Outg. I P2 | 2.83 |
| Outg. I P3 | 2.38 |
| Outg. I P4 | 2.38 |
| Outg. I P5 | 3.26 |
| Outg. I P6 | 3.11 |
| Outg. II P7 | 2.53 |
| Outg. II P8 | 2.55 |
| Outg. III P9 | 3.87 |
| Outg. III P10 | 2.46 |
| Distance P1 & P2 | 3.96 |
| Distance P2 & P3 | 2.53 |
| Distance P2 & P4 | 3.13 |
| Distance P1 & P5 | 3.61 |
| Distance P1 & P6 | 3.33 |
| Distance P7 & P8 | 2.77 |
| Distance P9 & P10 | 1.75 |
| Distance between outg. I & II | 20.22 |
| Estimation length outg. I | 20.1 |
| Estimation height outg. I | 8.9 |
| Estimation height outg. II | 8.0 |
| Estimation maximum inner diameter | 30.52 |
| Estimation maximum middle diameter | 40.68 |
| Estimation maximum outer diameter | 60.44 |
| Width ring before outg. II | 5.02 |
| Width ring before outg. I | 5.58 |

effects, as it punctuates the necklace with a blue/green note that differ from the dominant white and red.

At least four from the seven flat calcite beads were discovered in alignments *in situ*, next to the area of the upper chest and neck. The unique two tubular amber beads were found in similar front and upper positions, and one was found next to one of the spheric hematite bead. It is therefore tempting to think that the second (broken) amber bead was also placed next to second hematite bead.

The spherical hematite beads were discovered not far one from the other, on the left and right of the manubrium bone (*cf*. S1 Appendix). Beside the contrast created by the dark colour, the hematite beads found in a symmetrical position were likely employed to canalize or guide the rows of beads, that is, to start gathering them before reaching the buckle (Fig 22 right).

The size and the perforations of these beads are larger than any other type, and the rims and walls of perforations show intensive use-wear marks, although it is not possible to determine whether these traces were the result of one or several strings rubbing inside the perforations. These beads also show impact points and scratches all over the surfaces, which is to be expected due to their bigger size and roundness. Curiously, the calcite and the *Tridacna* beads do not present similar heavy wear despite the softness of their materials and being part of the same ornament. Irrespective of whether the hematite beads served the function of "row gatherer" or not, the presence of these unique items within such a complex necklace composed of

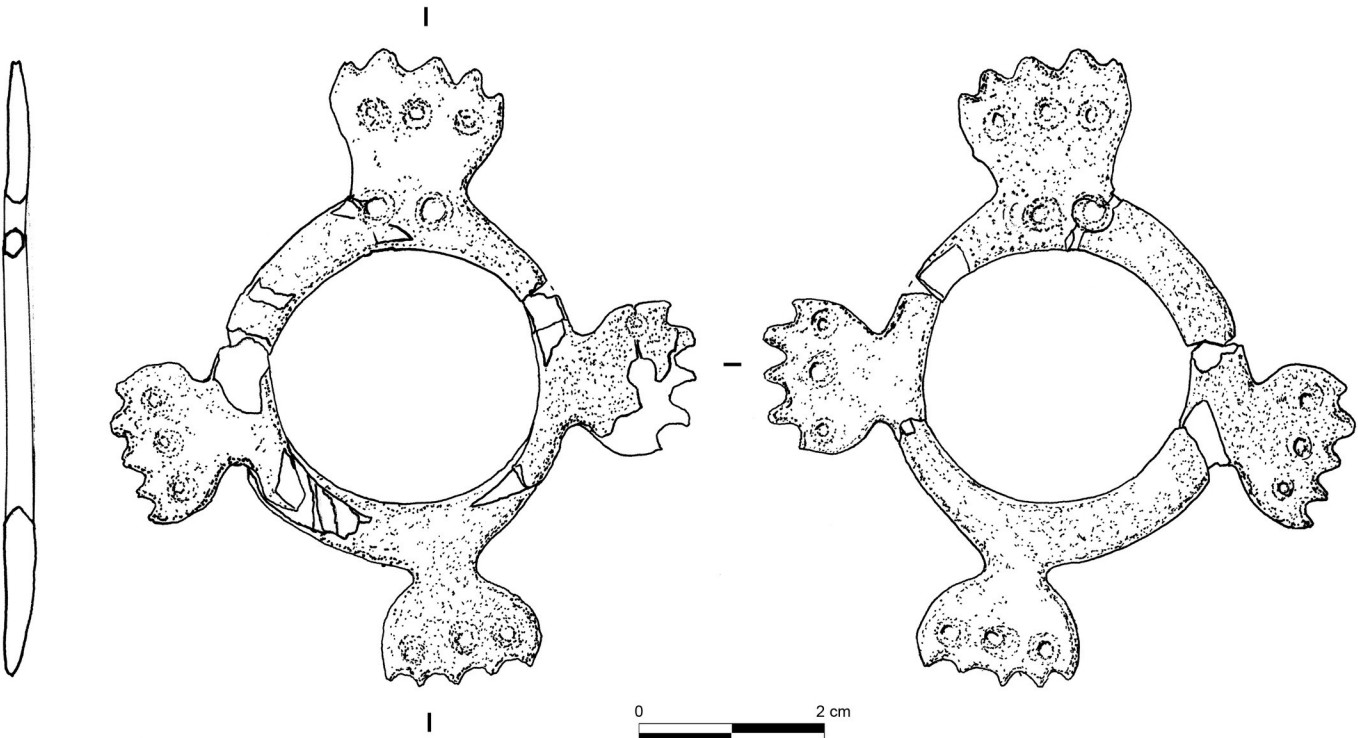

**Fig 19. A complete example of an engraved mother-of-pearl ring discovered in 2001 in the collective burial of Area D at Ba'ja.** Photo: H.G.K. Gebel; Drawing: B. Winkler. Our reconstruction resulted in quite sophisticated ring (Fig 20), with four symmetrical outgrowths decorated with multiples equidistant perforations.

many rows makes a lot of sense. Their placement between the shoulders and the manubrium bone would allow for adjusting the size of the necklace (for instance reducing the horizontal breadth of the rows). It is not possible to estimate how many strings were introduced inside these beads. At the same time, it is not necessary for the strings of several rows to pass inside the perforation, as the gathering of the rows can be made through simple discrete knots at the level of these beads. While the structure of the necklace that we propose uses the hematite beads to gather the exterior rows (R10 to R4) before reaching the buckle (Fig 22 right), a number of other possibilities can be imagined. For the physical reconstruction, we only gathered two rows in order to avoid damaging the beads, one of them being fragile.

The buckle found behind the neck of the child was made from the same dense and resistant hematite as the spherical beads. The intense use-wear that has affected the original shape of the perforation rims on this buckle indicate the tension exercised on each extremity. Hence, it is reasonable to think that the use of such material might have been precisely intended to serve certain functions that other softer beads cannot serve. On the contrary of the previous examples, the tiny four *Conus* disc beads and the small tusk shell portion can hardly provide a visual effect. Their use within the necklace is likely symbolic. Finally, it should be noted that the horizontal, oblique and longitudinal bands that ornament the *Tridacna* beads are not correlated to specific emplacement within the necklace. The occurrence of these patterns is most likely the result of a techno-economic processing of *Tridacna* shell fragments for bead making rather than an aesthetic choice.

The weight of the necklace when including the beads, the buckle, and the ring is estimated to an average of 192.23g, a minimum of 159.57g, and a maximum of 226.50g. The weight of

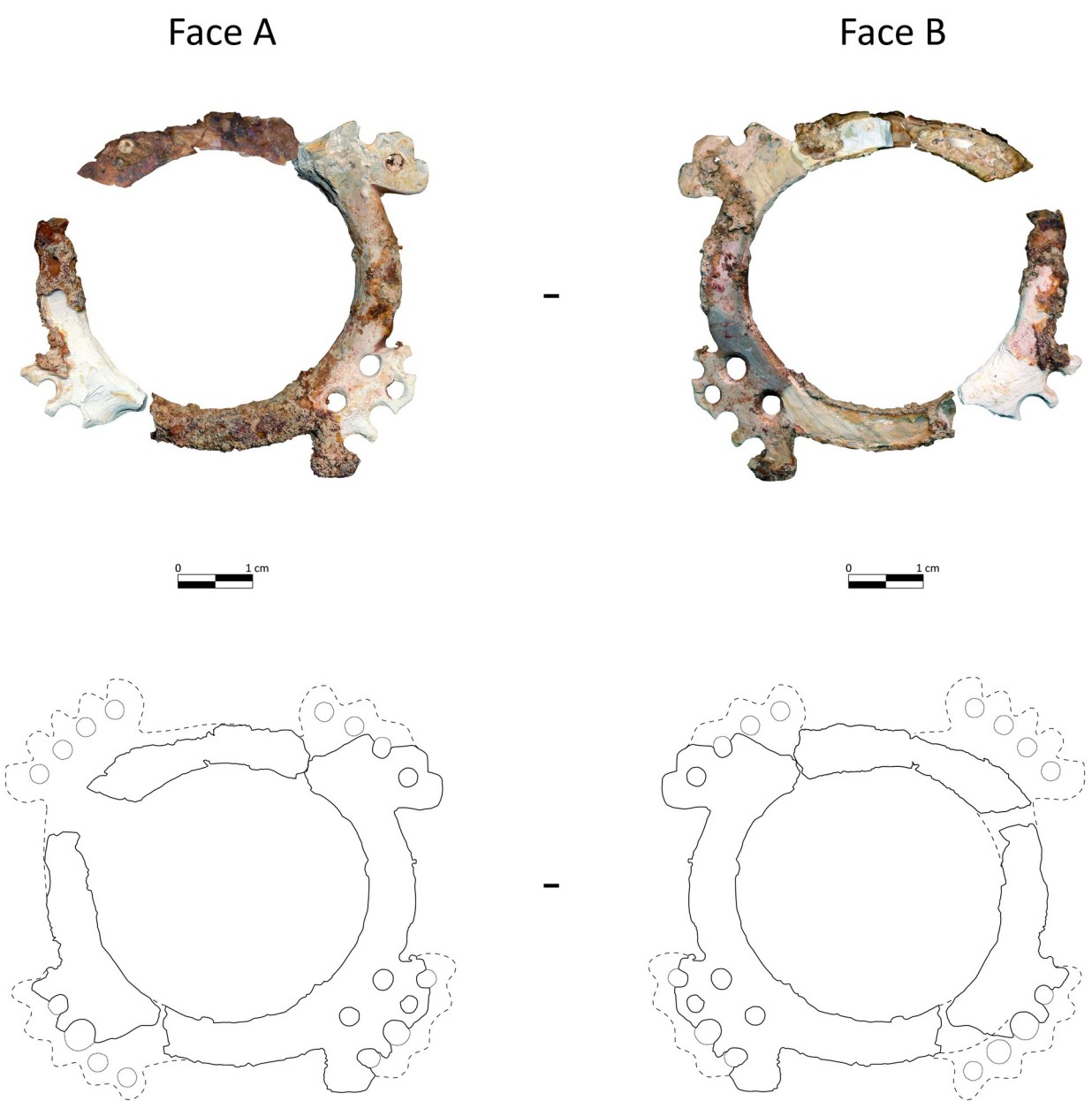

## Face A        Face B

**Fig 20. Hypothetical reconstruction of the complete shape of the mother-of-pearl ring found in burial CG7.** Scans and refitting: H. Alarashi and A. Burkhardt; restoration: A. Burkhardt, drawing of the hypothetical reconstruction: H. Alarashi, Baʻja N.P.

strings (totalling 6 m length) was not estimated as we lack information about their nature and whether or not they were treated by some substance which would influence their weight. The presence of beads showing residues of ochre inside the perforation prevent exclusion of the use of strings stained with red ochre, which might increase the weight of the necklace. In all cases, we assume that if the necklace was only composed of beaded strings, the ring and the buckle (excluding beads made from perishable materials), the total weight might not have reached 500g.

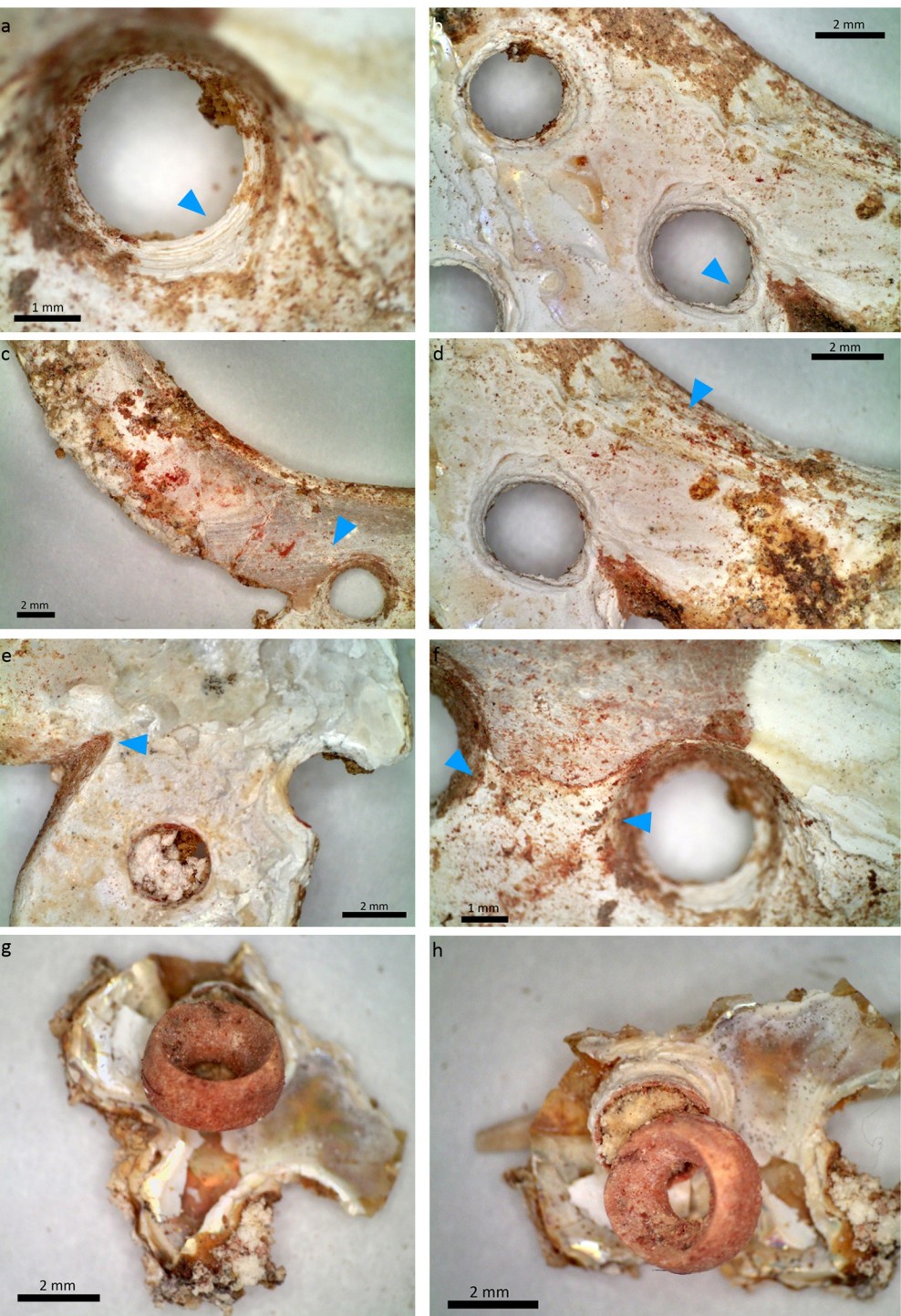

**Fig 21. Manufacture and use-wear traces observed on the mother-of-pearl ring.** A) Regular concentric striations inside the perforation. B) Deformation of the rim of the perforation (?) towards the external circumference of the ring. C) Parallel striations of abrasion (removal of the periostracum to reveal the nacre). D) Deep parallel groves made to extract the ring from the shell valve. E) Sawing/engraving marks made to create the outgrowths and provide the curved edges. F) Area between two perforations showing delimited depression and erosion of the nacreous layers (path of a string between these two perforations?). G-H) A disc bead stuck to a perforation of the ring with an unidentified mass (visible in picture H, remains of organic material, string? See also the obstructed perforation in picture E, showing similar material). Photos: H. Alarashi.

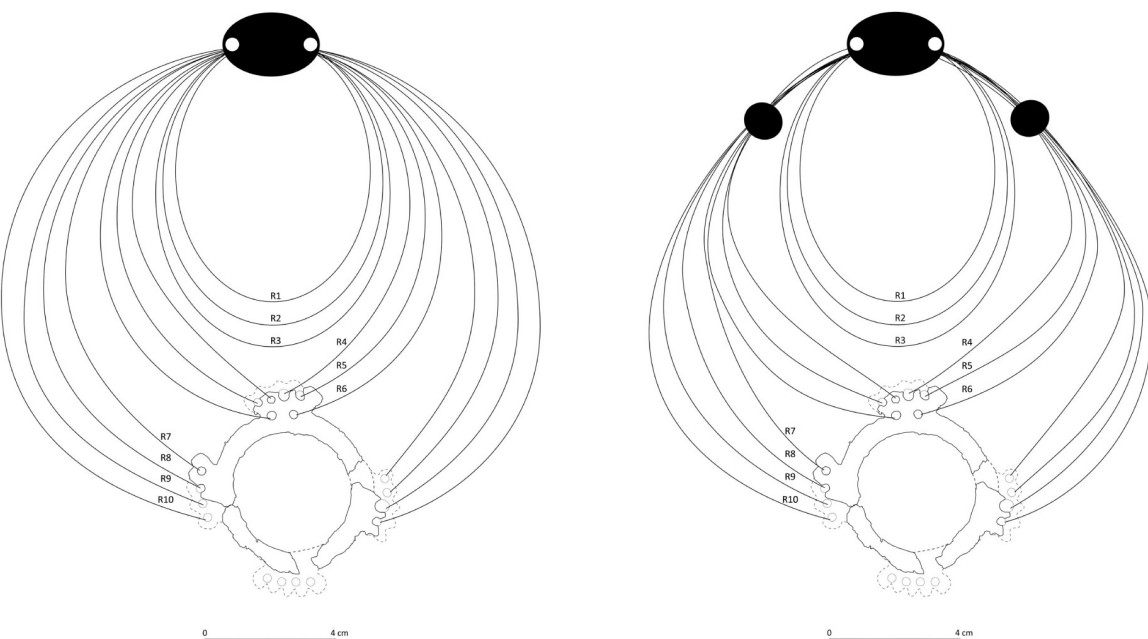

**Fig 22. Structure of the necklace between the ring and the buckle.** Left proposition does not include the black hematite beads. The right proposition displays the structure when the exterior rows (R4 to R10) are gathered with the spherical beads. Graph: H. Alarashi, Ba'ja N.P.

## Discussion

The reconstruction of the necklace we propose is based on field documentation obtained during the excavation, in-depth analysis of its constituent elements, and a series of estimations and logical arguments. Although our objective was to follow the simplest and most objective scenario possible, the reconstruction of the necklace turned out to be complex due to key items such as the ring and the large number of beads, among other factors. In other words, our proposition is one among others that are certainly not less complex. In the last few years, Ba'ja has yielded other spectacular ornaments discovered in children's graves. While preliminary results indicate that there is a great variability in the arrangement of beads with respect to the types, colors, and materials [6]. Consequently, the resulting ornaments appear to be as complex as the child's necklace. As for the necklace, reconstructing these other ornaments is crucial in revealing their diversity at Ba'ja, and in exploring their role in death rituals. Furthermore, revealing these compositions may provide insight into the cultural conventions that guided their conception. Specifically, it would enable exploring at which level body ornamentation was influenced by cultural norms; at the level of bead types, colors, or at the level of the whole ornament (necklace, bracelet, belt, diadem, etc.) and/or who was permitted to wear them? Answering these questions would open up promising avenues for understanding how the society of Ba'ja was structured and whether there were individual expressions in body ornamentation. The following discussion explores the significance of the necklace within the cultural context of the LPPNB and reveals new aspects regarding the early farming communities that are worth considerations.

### Beyond symbolism: A spectacular piece of art

Before delving or speculating on the meaning behind attributing this necklace to the deceased child, it is worth to contemplate its aesthetic qualities (Fig 23).

In the context of research developing social [83], semantic [84], psychologic and neurological [85, 86] approaches to understand the symbolic contents materialized by figurines, engravings, paintings or body ornaments, acknowledging the aesthetic tastes, beauty, radiance, or charm of creations might sound subjective and superficial. However, one should bear in mind that it is through the endless possibilities of material transformation that humans have developed genuine artistic abilities. Here we are not arguing that the artistic dimension of the necklace is autotelic, we do believe that body ornaments have social and cultural functions to accomplish through sets of information they communicate about their wearers (and makers). We are instead questioning, as lately formulated, whether "*aesthetic rules were symbolically guided, or directly perceived*", or if there were "*a third possibility, wherein perception and conception are brought together in yielding a significative form of art, otherwise seen as an artistic form of signification*" [87] (p. 6.) The aesthetic qualities of creations impact our senses and provoke emotions [88, 89].Consequently, emotions may play a role in formulating the meanings we attribute to these creations. In other words, the emotional response would be an influential factor that precedes the cultural codes.

The necklace, for instance, rivets our visual system and captures our attention, even when taken out of its informative Neolithic context. Its sheer volume when displayed draws our vision as it is indeed a very visible creation with limits spreading inside a 30x30 cm square. Symmetry, defined as "*a sense of harmonious, aesthetically pleasing proportionality that embodies beauty*" [90] (p. 1592), see also the definition of symmetry in Collins English Dictionary, 2023), was obtained through complex radial curvilinear structure that provides harmony [91] while allowing to explore the creation horizontally, through shapes, colors, and combinations of beads, and vertically, leading inevitably to the mother-of-pearl ring. The shape of the ring displays symmetry too. The distribution and size of its engraved outgrowths as well as the number of holes on each of them are balanced. The shape of beads and the bilaterality of the perforations of the buckle also respect symmetry, not to forget the intriguing divisible number of beads per type that allows almost perfectly even distributions within the rows.

Harmony is also about performing strategic distribution of the objects to reveal their beauty and qualities. While post-depositional alteration processes have given most beads a faded appearance, the analysis of the aesthetic nature of the adornment requires an imaginative effort to visualize the original brightness of each of the elements. In this perspective, it is not surprising that the masterpiece of the necklace was the ring, a large extremely delicate item that was skillfully engraved with complex fine motifs reminiscent of lace or filigree. The aesthetic sensibility is indisputable when appreciating how *Tridacna* beads reveal the beauty of the material and the effects of the alternation of the growth lines between mat and translucent bands. Shaped in different manners to show either horizontal, oblique or longitudinal patterns, these beads also show differences in the width of the bands. Their integration in all the rows not only dynamizes but also complexifies the necklace as it gives the impression that these beads have different morphologies despite having all the same shape. On the other hand, the combinations provide an appealing play of light and color. One should imagine the mother-of-pearl ring in its original state, before its degradation due to time and burial conditions, when it still possessed a shiny nacreous layer that reflected light and iridescenced in several colors at once [92], like nowadays mother-of-pearl jewelry and buttons. It is not surprising that this material was employed as inlays to simulate the eyes of the plastered skulls of the PPNB site of Yiftahel [93].

The milky and shiny surface of the ring contrasted with the dark brownish hematite objects, characterized by their metallic reflection [94]. While light may emanate from the mother-of-pearl and hematite objects, the best representation of color is demonstrated by the ubiquitous red calcite beads, the red being widely acknowledged as the most appealing and easily discernible color among others [95, 96]. Finally, this necklace has no parallel in any of the Neolithic

**Table 4. Estimations of the total length of the ornament.**

| Type | Measured beads (GN) | | | | | | Other beads (GS) | | | | Total Length (estimations) | | | |
|---|---|---|---|---|---|---|---|---|---|---|---|---|---|---|
| | Length | | | | | Total length | Estimated total length | | | | N | Mean | Range | |
| | N | Mean | s.d. | Min. | Max. | | N | Mean | Range (95%) | | | | (95%, mean ±2 s.d.) | |
| CDB cal. | 1492 | 1.42 | 0.347 | 0.58 | 3.67 | 2122.57 | 772 | 1098.27 | 1079.38 | 1117.17 | 2264 | 3220.84 | 3201.95 | 3239.74 |
| TB shell | 126 | 9.34 | 2.274 | 4.49 | 14.24 | 1177.18 | 106 | 990.33 | 944.44 | 1036.21 | 232 | 2167.51 | 2121.62 | 2213.39 |
| TB cal. | 46 | 7.26 | 1.751 | 4.63 | 10.25 | 334.00 | 20 | 145.22 | 129.87 | 160.56 | 66 | 479.22 | 463.87 | 494.56 |
| TB resin | 1 | 9.31 | 0.000 | 9.31 | 9.31 | 9.31 | 1 | 9.31 | - | - | 2 | 18.61 | 18.61 | 18.61 |
| Conus d. | 4 | 2.31 | 0.492 | 1.91 | 2.98 | 9.22 | 0 | - | - | - | 4 | 9.22 | 9.22 | 9.22 |
| ODB tur. | 5 | 2.16 | 0.575 | 1.43 | 2.82 | 10.78 | 0 | - | - | - | 5 | 10.78 | 10.78 | 10.78 |
| FB | 7 | 6.40 | 6.404 | 5.50 | 7.15 | 44.83 | 0 | - | - | - | 7 | 44.83 | 44.83 | 44.83 |
| DPP | 1 | 29.13 | 0.000 | 29.13 | 29.13 | 29.13 | 0 | - | - | - | 1 | 29.13 | 29.13 | 29.13 |
| SB hem. | 2 | 9.34 | 2.447 | 7.61 | 11.07 | 18.68 | 0 | - | - | - | 2 | 18.68 | 18.68 | 18.68 |
| Dental. | 1 | 6.12 | 0.000 | 6.12 | 6.12 | 6.12 | 0 | - | - | - | 1 | 6.12 | 6.12 | 6.12 |
| **TOTAL** | 1685 | | | | | 3761.82 | 899 | 2243.12 | 2202.24 | 2284.00 | 2584 | 6004.94 | 5924.81 | 6085.07 |
| Total length (in meters) | | | | | | | | | | | | 6.00 | 5.92 | 6.09 |

The estimation are based on the metric data obtained from the GN (Group of Nice) beads, and the estimated length calculated for the non-measured beads (GS: Group of Stuttgart). The estimations were first made per type of beads, then the results summed. C = circular; D = disc; B = bead; T = tubular; O = oval; F = flat; DPP = double perforated pendant; S = spherical; s.d. = standard deviation.

**Table 5. Estimations of the number of beads per type to be integrated according to the lengths of the rows (cm).**

| Separated rows | | CDB calcite | TB shell | TB calcite | Total |
|---|---|---|---|---|---|
| Row 1 | N | 100 | 24 | 10 | 134 |
| | Length | 142.19 | 224.22 | 71.88 | 438.29 |
| Row 2 | N | 110 | 24 | 10 | 144 |
| | Length | 156.41 | 224.22 | 71.88 | 452.51 |
| Row 3 | N | 110 | 26 | 10 | 146 |
| | Length | 156.41 | 242.91 | 71.88 | 471.20 |

| Connected rows | | Right side | | | | Left side | | | | Total | | | |
|---|---|---|---|---|---|---|---|---|---|---|---|---|---|
| | | CDB calcite | TB shell | TB calcite | Total | CDB calcite | TB shell | TB calcite | Total | CDB calcite | TB shell | TB calcite | Total |
| Row 4 connected | N | 76 | 13.5 | 3 | 92.5 | 76 | 13.5 | 3 | 92.5 | 152 | 27 | 6 | 185 |
| | Length | 108.12 | 126.13 | 21.563 | 255.81 | 108.12 | 126.13 | 21.563 | 255.81 | 216.24 | 252.25 | 43.125 | 511.62 |
| Row 5 connected | N | 91 | 12.5 | 3 | 106.5 | 91 | 12.5 | 3 | 106.5 | 182 | 25 | 6 | 213 |
| | Length | 129.46 | 116.78 | 21.563 | 267.81 | 129.46 | 116.78 | 21.563 | 267.81 | 258.92 | 233.57 | 43.125 | 535.61 |
| Row 6 connected | N | 111 | 12 | 2.5 | 125.5 | 111 | 12 | 2.5 | 125.5 | 222 | 24 | 5 | 251 |
| | Length | 157.91 | 112.11 | 17.969 | 287.99 | 157.91 | 112.11 | 17.969 | 287.99 | 315.82 | 224.22 | 35.938 | 575.99 |
| Row 7 connected | N | 136 | 10.5 | 2.5 | 149 | 136 | 10.5 | 2.5 | 149 | 272 | 21 | 5 | 298 |
| | Length | 193.48 | 93.427 | 17.969 | 304.87 | 193.48 | 93.427 | 17.969 | 304.87 | 386.96 | 186.85 | 35.938 | 609.75 |
| Row 8 connected | N | 156 | 10.5 | 2.5 | 169 | 156 | 10.5 | 2.5 | 169 | 312 | 21 | 5 | 338 |
| | Length | 221.93 | 93.427 | 17.969 | 333.33 | 221.93 | 93.427 | 17.969 | 333.33 | 443.86 | 186.85 | 35.938 | 666.65 |
| Row 9 connected | N | 186 | 10 | 2.5 | 198.5 | 186 | 10 | 2.5 | 198.5 | 372 | 20 | 5 | 397 |
| | Length | 264.61 | 93.427 | 17.969 | 376.01 | 264.61 | 93.427 | 17.969 | 376.01 | 529.22 | 186.85 | 35.938 | 752.01 |
| Row 10 connected | N | 216 | 10 | 2 | 228 | 216 | 10 | 2 | 228 | 432 | 20 | 4 | 456 |
| | Length | 307.29 | 93.427 | 14.375 | 415.09 | 307.29 | 93.427 | 14.375 | 415.09 | 614.58 | 186.85 | 28.75 | 830.18 |
| L total necklace | | 1599.7 | 1074.4 | 237.19 | 2911.3 | 1599.7 | 1074.4 | 237.19 | 2911.3 | 3199.3 | 2148.8 | 474.38 | **5822.5** |

C = circular, D = disc, B = bead, T = tubular, O = oval, F = flat, DPP = double perforated pendant, S = spherical.

Levantine ornamental traditions known thus far. The large volume, complex organization, symmetry, harmony, beauty of objects, play of lights and colors are in fact reminiscent of the refined ornaments of the latter urban Mesopotamian and Egyptian societies [97, 98].

## Wealth and other intangible values

The necklace has an undeniable appeal, a glimpse is enough to produce the attraction and contemplation of its attributes. Such a stimulation of curiosity is likely to generate knowledge. In light of other ornaments from Ba'ja, we suggest that the assembled objects were not selected randomly. The choices of materials, colors, types, and morphologies were meaningful, not only to serve the design and the volume (see above) but also to express intangible values, the most obvious and defendable of them being those of abundancy, diversity, and exclusivity.

The abundance of beads composing the necklace, which is a common trait of ornaments found in other burials at Ba'ja e.g. [6, 58], hints to wealth and prosperity. While it is impossible to decipher the symbolic code (growth, fertility, community?) behind abundance, the mere intention to assemble as many as beads as possible, of non-utilitarian yet sought-after items, does express material wealth and access to valuable resources. By material wealth we mean "*tangible possessions of value*" [99] (p. 295 and references therein). There is no better example to embody this definition as ornaments. Accessibility to wealth is even more strongly signified when, as in the case of the child's necklace, the ornament is definitely extracted from life circles and deposited in sealed graves. Burying valuable objects such as body ornaments with the deceased may reflect an economic strategy that aims at controlling supplies, thereby at maintaining beads value within living populations. Withdrawing beads from life circulation by hiding or burying them is in fact a universal and intemporal practice, that has been often attested by the discovery of hoards of precious objects [100–102] (usually forgotten by their owners). Some ethnographic examples may illustrate metaphorically the economic interest for beads. For instance, some Krobo people in Ghana believe that specific beads, the old-African powder-glass beads, have spirit and will reproduce in the ground if buried [103]. For other traditional populations such as the Tani and Naga tribes in Northeast India, the sought-after necklaces of glass and stone beads are transmitted over generations as heirloom, and only highly respected individuals are buried with ornaments [104]. Whether the necklace was intended to accomplish an economic objective, to express the esteem to the person of the child, or to provide wealth (or paraphernalia) for the afterlife domain remains an open question. However, we can safely suggest that at least part of the people of Ba'ja had ensured access to prestigious and valued exotic materials, even if they were later transferred into other realms, beyond life.

Adornments with a large number of beads (over 2500) are unprecedented among contemporary Neolithic villages in the Levant. However, they appear more frequent in contexts of hunter-gatherers. Examples of these date from early periods such as the exceptional ornaments found in Upper Palaeolithic burials at Sunghir in Russia [105], those from the Early Natufian culture in Southern Levant [106], and even from the early Neolithic in the Upper Tigris Valley [44]. The lack of similarly abundant beaded ornaments from other farming Levantine known villages may be due to cultural choices where other values prevail over abundance. Preservation issues might also be responsible. Ornamental items were certainly not limited to those that have survived the ravages of time, but likely included organic materials such as seeds, wood, animal skins, feathers, and others (see for example [107, 108]), thereby enhancing this impression of abundancy. On the other hand, the opulent character of the necklace reflects diversity and exclusivity, two interdependent and complementary values. Diversity is defined here by the union of elements serving the same purpose (the necklace) yet exhibiting different

visual aspects due to natures or to modifications of their materials. Exclusivity refers to the rare items among the whole group, those limited in number and, thus, not available to every-one. The design of the necklace has precisely succeeded in balancing diversity and exclusivity. While the former requires multiplicity that may emphasize the latter, the latter relies on the singularity that enriches the former. In addition to substantially fulfilling the aesthetic function of the necklace, these values reflect the knowledge, the economic investment, the extended cultural connexions and relationships of at least part of the inhabitants of Ba'ja.

## Gathering regions, connecting cultures

We identified eight different materials that were transformed, according to our typological classification, into four functional categories and seven morphotypes. In terms of materials, the diversity is illustrated by the use of shells from four mollusc genera (belonging to three different classes: gastropods, bivalves and scaphopods), three types of stones, and a hitherto unprecedentedly identified fossil amber.

In the case of shells, at least two species are originally from the Red Sea: *Pinctada margaritifera* and *Tridacna* sp. The first provides the nacreous shiny mother-of-pearl while the second offers a thick whitish material exhibiting the growth lines in an alternating opaque and translucent pattern. These shells were exploited as raw materials, which means that their quality was an important criterion of selection, a requirement that is not always guaranteed when gathering shored shells after the death of the animal, as they are exposed to natural damage of sand erosion, weathering, rolling, impacts, etc. [109, 110]. In the specific case of *Pinctada*, the size of the shell was also important to extract large rings as those discovered in Ba'ja, some measuring over 10 cm [73] fig. 5t, or as the necklace's ring. In fact, the outermost diameter of the latter, including the external edges of the outgrowths, reaches over 6 cm. However, one should consider that the valve from which this ring was extracted should have been not only large enough to provide a 6 cm diameter, but also to choose the area of the shell where it is sufficiently flat. This makes an estimation of the maximum diameter of *Pinctada* valve used for the ring of at least 7 to 9 cm. It is worth mentioning that a *Pinctada* valve shaped into a relatively flat pendant (with two small perforations), with a diameter of over 15 cm was discovered at Basta, meaning that the complete valve was even larger. Considering the size and the quality of the mother-of-pearl, it is hard to defend an acquisition of *Pinctada* valves through random gathering from shored specimens on the beach. Thus, we suggest that large valves were rather collected from their natural habitat. One should add that access to these molluscs might have been relatively easy as *Pinctada margaritifera* (and *Tridacna* (*maxima*?) live in shallow, and warm, waters of the Red Sea and the Gulf of Aqaba [111–114]. In fact, their over-exploitation and extension in many of their habitat regions today is partly explained by their accessibility [115]. If they were procured from their natural habitat during the Neolithic, one should consider the awareness about this aquatic source, and the know-how related to fishing, cleaning and processing of the material.

Three varieties of stones, calcite, hematite, and turquoise were used for the necklace. The red calcite is dominant counting over 99% for the group of stone beads. Physicochemical analysis indicates a quite pure variety of calcite with a $CaCO_3$ ratio over 91%, with enough presence of $Fe_2O_3$ to give the red color. The occurrence of calcite according to this high ratio is so far unknown for the area of Ba'ja. Instead, several sources of the sedimentary limestone rocks are available (less than 5 km to the north and east of Ba'ja, [116]). There is a mention of calcite in the Greater Petra Area, between the two contemporaneous sites of Ba'ja and Basta (Zone 1 according to [117]), yet without precision regarding its color or chemical composition. Red calcite can result from calcite precipitation within sedimentary rocks rich with iron. This

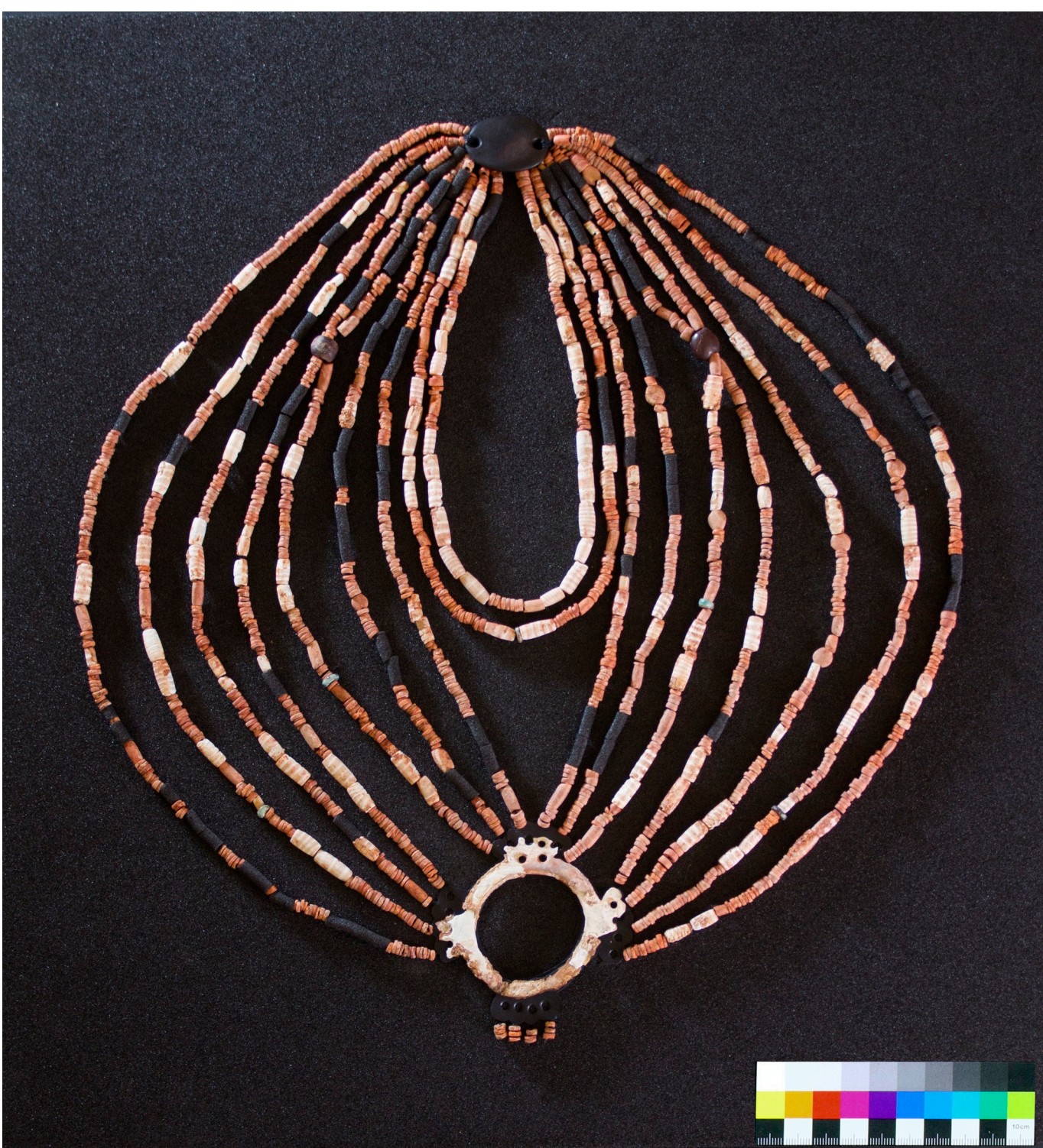

**Fig 23. Final physical reconstruction of the necklace, today exposed at the new museum of Petra in Jordan.** Note that the use of black foam tubes was intended to substitute the *Tridacna* and calcite beads which were too poorly preserved to be displayed. For preservation and museology reasons, the physical reconstruction did follow exactly the reconstruction proposed in Fig 22 right. For example, we avoided the path of more than two strings in the perforations of the hematite beads as one of them was fragile. Photo: A. Burkhardt, Drawings: H. Alarashi, Ba'ja N.P.

phenomenon is documented for the Transdanubian Range in Hungary [118] and can be envisaged for the Greater Petra Area when taking into account that this region is part of the Amman-Wadi Es Sir formation (B2/A7), which is a karstic system of silicified limestone [119], fig. 1. That is, red calcite might occur in the region and it potential sources, possibly underground (?), may have been known and exploited by the Neolithic communities.

Hematite is a very widespread mineral and occurs in many regions of Southern Levant including the Greater Petra Area [117, 120]. It is worth mentioning that iron-based oxides provide red ochre, which is very common at the site and was found in grave CG7, next to the legs of the child. Although we don't precisely know their chemical composition, the hematite stones and the ochre might pertain to the same geological formations. Whether the red calcite, the hematite, and the ochre were obtained from the immediate vicinity of Ba'ja or from sources located far away Is still an open question. However, one should keep in mind that Ba'ja is located in a very rich mineral environment that was exploited for building the dense and complex household units, and for the lithic industries based on a wide range of raw materials such as flint, sandstone, limestone, clay, quartz, quartzite, and marl [58, 116, 121, 122]. It is therefore worth raising the question regarding the exploitation of local resources for bead making activities at Ba'ja.

Turquoise has a narrow distribution in the Levant being limited to mines located in Sinai, at Wadi Magarah, Gebel Adeida and Serabit el-Khadim [123, 124]. Although direct historical (epigraphic) evidence of turquoise mining from the Sinai sources are numerous [125], the procurement of the necklace turquoise from these same sources has yet to be confirmed through ongoing geochemical and isotopic [126] characterization of both geological and archaeological samples. Meanwhile, two other potential sources of turquoise should be noted. The first is recorded for the copper mines of Timna in the Negev desert by Mindat.org (https://www.mindat.org/loc-2066.html), the mineral database of Hudson Institute of Minerology, while the second, mentioned by Hauptmann [123], corresponds to turquoise deposits to the East of the Red Sea, in the Tabuk region, Northwest of Arabia Saudi. Whatever was the geological source, the turquoise used for the necklace pertains to an allochthonous origin, meaning that the owners had to invest in one way or another to acquire this valuable material either as raw fragments or as finished or semi-finished beads.

Diversity in material choices is finally represented by the use of two fossil amber beads. Badly preserved, it should not be excluded that other amber beads were part of the necklace, although fragments of destroyed beads would probably have been discovered as the excavation was extremely meticulous and sediments systematically sifted. The spectra provided by the FTIR analyses of the two beads are compatible with the features recorded for the amber occurring in Lebanon [71, 72]. In fact, the Near East has the oldest amber deposit known worldwide [127], the so-called "Levantine amber belt" [128] dating to the Lower Cretaceous [129]. Extending over 200 km from Southern Lebanon to the Negev desert, several deposits in sandstone and silt formations were recorded. Comparisons between the spectra of the beads and those of the geological references are therefore needed in order to identify the area and distance of the source exploited during the Neolithic. The closest amber spots might correspond to the Barboor deposits (https://www.mindat.org/loc-190761.html) located at least 60 km (up to 110) Northwest of Ba'ja, on the Western side of the Jordan Valley.

The necklace represents a unique and singular entity created through the combination of thousands of entirely shaped elements (except for five tiny shells that might have been modified naturally). In terms of morphotypes, all the beads pertain to a repertoire fundamentally Neolithic, coherent with the LPPNB ornamental spectra e.g. [130–135]. Although certain functional types have appeared in earlier Epi-Paleolithic times [73] such as disc beads and double perforated pendants, attested for the final Natufian in the Levant [81, 132, 134], long

perforations drilled in compact dense materials such as stones and shell bulks to create long beads appears to be a Neolithic innovation (Alarashi et al. submitted). Long beads have developed over time into several typological families and tens of types declined into geometric, anatomic (natural shapes or imitations) and singular (abstract, complex) forms [134]. The creators of the necklace show a clear predilection for beads with rounded geometric forms represented through disc, tubular, flat and compact beads. In fact, these are all the typo-functional families of beads that have been developed in the Levant during the Neolithic [134], and they are all represented within the necklace. The integration of the flat beads within the necklace is very interesting. Here again, the low number and small size of these beads are subtle indications of intangible values. Indeed, flat beads with biconvex section represents the hallmark of the PPNB cultures in Anatolia, the Middle and Upper Euphrates Valley and central Levant [136]. They occur in the Neolithic villages of Aşıklı, Çayönü, Mezraa Teleilat, Tell Halula, Abu Hureyra, Tell Aswad and Tell Ramad (Fig 1), among others [136–139]. In these sites, these beads have impressive aspects due to their large sizes and shiny colorful hard materials. Made from carnelian, agate, amazonite, turquoise, obsidian and other semi-precious stones, the so-called "butterfly" beads represent the earliest clear examples of specialized stone bead crafts [139, 140] and high-valued prestigious items of the LPPNB cultures. Although made on softer materials, red calcite (in the case of the necklace) and *Tridacna* shells [73], the flat beads of Baʻja mimic the geometric forms known from the North, in particular the oval and diamond shapes. The use of flat beads for the child's necklace suggests therefore a cultural interaction with other farming communities in the Levant. While this type can be created separately and independently from septentrional influences, indications of intensive contacts between Southern and Northern Levant during the period of occupation of Baʻja are provided precisely by body ornaments found in LPPNB sites of the Middle Euphrates such as Tell Halula and Abu Hureyra, namely cowry and *Nerita* shells from the Red Sea and turquoise, copper ores and probably amazonite from Sinai and southern Levantine mines [134, 141, 142]. Besides the exotism of some materials used for the necklace, the integration of specific types such as the flat beads may suggest affiliation to a broad network of farming communities in the Levant.

By providing these examples of diversity and exotism of materials and types used for the necklace, the aim is also to highlight the level of connectivity between the people of Baʻja and the wider world. Despite its location (Fig 1), invisible and hidden between the rocks, the village had access to marine, mineral, and amber sources, resulting in a diverse range of ornamental designs. The topography of the village seems irrelevant in light of the fact that the people of Baʻja had very rich and diverse ornamentations that were not limited to funerary purposes but also to life as beads, pendants and rings were also found in non-funerary contexts [80].

## Making and sharing memory

The multicultural and multiregional features of the necklace are greatly emphasized by the local imprint materialized by the decorated mother-of-pearl ring and the attractive banded-patterned *Tridacna* beads. These elements are typical of the LPPNB culture to which pertain the communities of Baʻja and Basta in the Greater Petra region and, to our knowledge, they do not occur elsewhere in the Levant. A total of three specimens of *Tridacna maxima* (one modified) is documented at Shkarat Msaied [143]. Three others are mentioned within the malacological assemblage of Ayn Abu Nukhayla, more to the south in Wadi Rum [135]. That is, no indications of any of these types thus far.

The *Pinctada margaritifera* shell from which the mother-of-pearl was extracted occurs as almost complete valves, valve fragments, worked slabs and debris in all the excavated sectors of the site [80]. The technological analyses of the mother-of-pearl items at Baʻja show that the

production was mainly oriented towards the creation of large rings with clear technoeconomic management of the raw material [73]. That is, after the ring extraction from the mother-of-pearl slabs, the remaining fragments were transformed into small rings, disc beads, pendants, buttons, and other "free-style" items. These mother-of-pearl "by-products" follow generally the shape and the size of the "waste" fragments, revealing in some cases the artistic creativity of the craftspeople. The unworked fragments and fractured rings were kept for further transformation or for recycling. The predilection for large *Pinctada* valves, the technoeconomic management of the production and the duality of standard/atypical products suggest a form of mother-of-pearl industry at Ba'ja. In fact, sequences of the whole biography of the items–procurement, production, use, and deposition (or export?)–can be inferred, thus providing insights on how closely the technoeconomic, artistic and symbolic systems were interwoven.

If the masterpiece of the necklace, the decorated ring, was produced at Ba'ja, it is reasonable to think that at least, the assembling of the necklace took place there too. In this sense, it is worth asking whether the manufacture of the other elements of the necklace, the beads and the boucle, were also made at the village. While production areas with concentrations of raw materials, preforms, unfinished beads, debris along with the associated tools represent optimal contexts to interpret bead-making activities, indirect evidence should not be neglected. Aside from a few exceptional discoveries [41, 144, 145] or in some seasonal campsites established within specific mineral landscapes exploitable for bead-making [74], identifying bead workshops in densely constructed prehistoric settlements is a challenging task, especially if these craft activities did not take place in dedicated areas; separated from the domestic or collective units. One should consider that living in, or using structures was likely subject to cultural attitudes–such as cleaning, re-arranging, displacing, maintaining, etc. [146], which would challenge the interpretation of the archaeological observations. Moreover, raw materials and working tools are valuable and could be also fragile, especially the piercing and drilling ones. In densely populated areas, these items would have been retrieved immediately after their use [79]. Finally, most of the bead-making technologies, except for knappable varieties (quartz, carnelian, or amazonite), apply slow and soft technical operations such as abrasion, polishing, sawing, and drilling. Their debris consists of particles of abraded materials that are undetectable to the naked eye during excavation.

At Ba'ja, ornamental objects were discovered also in non-funerary contexts [73, 80]. About 14% of this assemblage was classified into the category of "technical objects". These include mineral and shell raw materials, unfinished stone and shell beads, and mother-of-pearl debris and broken items [73]. Apart from fragments of hematite, chrysocolla, carnelian and other undetermined stones, none of the minerals identified for the necklace were found as raw or as unfinished items at the site so far. From a technical point of view, the turquoise beads of the necklace, but also those studied from other burials [59, 77] and non-funerary contexts [73], were different in terms of shaping and finishing qualities compared to beads made from calcite, hematite, and *Tridacna* shell. Unlike the calcite, the *Tridacna* and the hematite items, the turquoise beads have irregular and non-standardized shapes. This suggests that the material, its exotism and physical proprieties such as its hardness, color or the aspect of the surface, prevailed the shape. It should be noted that providing a regular shape means reducing the amount of the material. In this sense, the irregularity of the beads is comprehensible (justifiable). Turquoise beads may have been imported as final (or semi-final) products to Ba'ja. The high intensity of use does not concern only the beads of the necklace but almost all the turquoise beads discovered at the site. Indeed, turquoise beads were used, reused and recycled [59], thus pinpointing their important value to the people of Ba'ja.

There is no direct evidence of hematite bead manufacturer. However, raw fragments are available at the site as its procurement could be made from the vicinity. This material was used

to create two different types, exclusive to the child's ornament: a double perforated pendant and compact almost spherical beads. The red calcite also occurs in different typological families, disc, tubular, and flat beads, which share identical technical traces. It is worth remembering that the calcite tubular and disc beads have similar diameter range. The range of length of the calcite flat beads also share similar values with the calcite tubular beads. Beside size standardization, which is mostly observed for the disc beads, there is a general size harmonisation between the three typological families. On the other hand, the calcite beads show very homogenous qualities in terms of regularity and symmetry of the shapes and surface treatment. Being shaped according to specific metric and qualitative standards, the calcite beads of the necklace appear as the result of a specific production unity applying a standardized method, or as the result of the work of a particular bead-maker, dedicated to the production of thousands of elements composing the bulk of the necklace composition.

*Tridacna* tubular beads show similarities with the qualitative features (regularity, smooth finished surfaces. . .) characteristic of the calcite beads. At Ba'ja, *Tridacna* shell beads do not occur only as tubular but also as flat oval and diamond-shaped types. Tubular beads in cylindrical and barrel shapes were preferred for the child's necklace, each featuring unique patterns of alternating natural bands (growth lines). The analysis of these patterns in relation to the lengths of the beads revealed an optimisation strategy in the exploitation of *Tridacna* valves. This reminds the technoeconomic management of mother-of-pearl items, which consists in transforming the fragments remained after the extraction of rings from the valves into diverse objects (buttons, small rings, pendants with double or three perforations, etc.). In sum, the production of objects from *Tridacna* and *Pinctada* valves was subject to a control of optimization of the material originated from faraway Red Sea source(s).

A local production of *Tridacna* beads is a plausible hypothesis supported, on the one hand by the fact that the distribution of these items seems thus far limited to the villages of Ba'ja and Basta, on the other, by the discovery of a *Tridacna* workshop at Basta [121] and one unworked fragment at Ba'ja.

The technological characteristics of the stone and shell beads used for the necklace and the ability of creating diverse challenging morphotypes from the same materials reflect knowledge, high skills, and great sensibilities, especially to manufacture and manipulate the tiny disc beads. To this should be added the intense investment in time and energy during the manufacture. Over two-thousands disc beads were integrated into the necklace rows. Although batch polishing accelerates the finishing stage when compared to a single-bead treatment [147], the previous stages of shaping and perforation stages remain considerably time-consuming.

Use-wear analyses of stone and shell beads from the necklace revealed differential patterns of use-intensity. Unexpectedly, the relatively soft calcite and the *Tridacna* beads only show superficial, random marks that were likely produced due to bead-to-bead alignment on the rows of the necklace. In contrast, the hematite and turquoise beads, which are dense, hard, and more resistant, display intensive wear with deep rounding of the edges, shiny polish, and slight deformation of the perforation rims. This means that beads of different use-wear intensities were combined to compose the necklace. The main structuring element of the necklace, the mother-of-pearl ring, was most likely created at Ba'ja, based on the fact that mother-of-pearl craft is attested at the site [73]. With this in mind, it is reasonable to think that the composition of the necklace, and probably its conception in terms of structure, number of beads, lengths of rows, etc., took place at Ba'ja too. This leads us to think that the bulk of items needed to furnish the rows and fulfil the general volume of the necklace, namely the red calcite and the *Tridacna* beads, were locally produced, at least within the Greater Petra Area, if not at Ba'ja itself. The intensively worn turquoise and hematite items were undoubtedly integrated into the necklace after having been used, probably by other individuals (e.g. elders of the child?).

Considering the production and use of the beads raises the question of whether the necklace existed during the lifetime of the child or whether it was created for the death occasion. While one can speculate on different scenarios, we prefer drawing upon the contextual funerary data to address this question. The significant technoeconomic investment and energy invested in the construction of the exceptional cist burial, as well as the funerary staging of the child's body and the accompanying ritual practices surrounding their death (Benz et al., in press; see also [58] for the detailed description of the burying events) clearly indicate the importance of this eight-year-old individual. When looking at the whole picture, the expression of the child's social status and identity appears as significant as the performance of the funerary act, and in both cases, the necklace played a determinate role. The lavishly endowed child laying down in an impressive burial, and the extraordinary ornament decorating the chest was intended to be shown, probably for a last sight. In this sense, the death of the child should be seen as a public event gathering the people of Ba'ja, families, friends, and probably members from other villages too. The shared experience, emotions, gratitude, or sorrow at this occasion certainly contributed to consolidating the community, in densifying the collective memory which are fundamental in such moments of loss (Benz et al., in press).

## Conclusion

The analysis of the child's necklace has yielded valuable information that enhances our understanding of the ritual practices and symbolic behaviour of the community of Ba'ja while shedding light on the artisanal and economic capabilities employed to serve these expressions. The study has also revealed an unexpected level of connectivity between Ba'ja and the wider world, and its involvement in the exchange and trade networks that circulated throughout the Levant during the LPPNB. Our in-depth analysis of the assemblage has allowed us to reimagine one of the oldest and most impressive Neolithic ornaments, believed to have been created to endow a highly distinguished 8-year-old child of the community. Despite its elaborate design, such a necklace was not created for exchange or trade purposes but was rather a part of the chil"s burial, serving as a significant testament to the cultural practices of the time. The ex- or de-commodification [148, 149] of the necklace during the burial ritual is remarkable. Its association with a child reaffirms the significant place of children for early farming communities in the Levant and questions their indirect role in stimulating bead production activities, in boosting technologies and aesthetics tastes, especially given that they appear to be the main individuals concerned by body ornamentations in such contexts (for sites with burials were body ornaments are more related to children, see e.g. [45, 57, 134, 136, 139, 150–157]). The study of the necklace reveals, on the other hand, how complex the interactions between the social actors of the community of Ba'ja had been–the bead-makers, the string/ cordage makers, the travelers, or mobile individuals, the familial or tribal authorities behind the demands of artistic creations, and other members of the society. In this, the necklace also reflects a complex narrative of the social dynamics within and outside Ba'ja that certainly merit further exploration.

## Supporting information

**S1 Appendix. The excavation: Method, conditions, observations and first hypothesis.** PDF document with text and four figures.
(PDF)

**S2 Appendix. Details on the applied methods.** PDF document of four parts: Palaeoproteomic (Part I), Infrared spectroscopy (Part II), CT-scan (Part III), Morphometric specificities for the

study of disc beads (part IV).
(PDF)

**S3 Appendix. Considered elements for the reconstruction of the ornament.** PDF with text, four tables and two figures.
(PDF)

## Acknowledgments

For the 2016–2019 excavations we acknowledge the support of H.E. Prof. Monther Jamhawi, then Director-General of the Department of Antiquities (DoA), Amman, and H.E. Dr. Yazeed Elayan, present Director-General of the DoA, Director of Excavations and Surveys, Dr. Aktham Oweidi and Dr. Ahmad Lash, Head of Loan Sector. Sincere thanks are due to the former President of the Yarmouk University, Prof. Zeidan Kafafi, to Prof. Hani Hayajneh, then Dean of the Faculty of Archaeology and Anthropology at Yarmouk University and Dr. Hussein al Sababha, Mousa Serbil, Julia Graf, and Martin Bader for their cooperation in the rescue of Burial C1: 46. We are grateful to Prof. Dr. Dominik Bonatz, Head of the Institute for Near Eastern Archaeology, Free University, Berlin for the administrative support. We acknowledge the logistic support of the laboratory of CEPAM (CNRS, University Côte d'Azur, Nice), the IMF-CSIC and the group of Archaeology of Social Dynamics and the team of the new museum of Petra in Jordan. Our thanks go also to A. Colonese from the University of Barcelona, to G. Monge, CEMEF laboratory of Sophia Antipolis, to Pauline Garberi from CEPAM and to Julia Schultz from State Academy of Stuttgart for their different insights and helps. We do not forget also Barbara Puskás (@ORF/@ARTE) and her filming team of the documentary "The mysterious Stone Age village". Finally, we sincerely acknowledge ex oriente e.V. association and the Ba'ja Neolithic Project that made this discovery and this research possible.

## Author Contributions

**Conceptualization:** Hala Alarashi.

**Data curation:** Hala Alarashi, Marion Benz, Julia Gresky, Jorune Sakalauskaite, Hans Georg K. Gebel.

**Formal analysis:** Hala Alarashi, Julia Gresky, Lionel Gourichon, Melissa Gerlitzki, Jorune Sakalauskaite, Beatrice Demarchi, Meaghan Mackie, Carlos P. Odriozola, José Ángel Garrido Cordero, Luisa Vigorelli.

**Funding acquisition:** Hala Alarashi, Marion Benz, Carlos P. Odriozola, Hans Georg K. Gebel.

**Investigation:** Hala Alarashi, Marion Benz, Lionel Gourichon, Hans Georg K. Gebel.

**Methodology:** Hala Alarashi, Alice Burkhardt, Andrea Fischer, Lionel Gourichon, Melissa Gerlitzki, Jorune Sakalauskaite, Carlos P. Odriozola, Miguel Ángel Avilés.

**Project administration:** Hala Alarashi, Marion Benz, Hans Georg K. Gebel.

**Resources:** Marion Benz, Andrea Fischer, Lionel Gourichon, Beatrice Demarchi, Meaghan Mackie, Matthew Collins, José Ángel Garrido Cordero, Miguel Ángel Avilés, Alessandro Re, Hans Georg K. Gebel.

**Supervision:** Hala Alarashi, Marion Benz, Hans Georg K. Gebel.

**Validation:** Hala Alarashi, Marion Benz, Julia Gresky, Andrea Fischer, Lionel Gourichon, Jorune Sakalauskaite, Beatrice Demarchi, Matthew Collins, Alessandro Re, Hans Georg K. Gebel.

**Visualization:** Hala Alarashi, Marion Benz, Alice Burkhardt, Lionel Gourichon, Jorune Sakalauskaite, Carlos P. Odriozola, Luisa Vigorelli, Alessandro Re.

**Writing – original draft:** Hala Alarashi.

**Writing – review & editing:** Hala Alarashi, Marion Benz, Lionel Gourichon, Beatrice Demarchi, Meaghan Mackie, Hans Georg K. Gebel.

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
