## [Decision Letter · Decision Letter 0]

8 May 2023

PONE-D-23-09318Adorning the dead child at the Neolithic village of Ba`ja (Jordan): A reassessment of the techno-symbolic behavior of early farming communities in the LevantPLOS ONE

Dear Dr. Alarashi,

Thank you for submitting your manuscript to PLOS ONE. After careful consideration, we feel that it has merit but does not fully meet PLOS ONE’s publication criteria as it currently stands. Therefore, we invite you to submit a revised version of the manuscript that addresses the points raised during the review process.

Both Reviewers 1 and 2 provide substantial comments on content and approaches. In light of these reviews, the manuscript needs some revision. Please consider each of these comments – especially the ones relating to the (economic) implications of the removal of beads from circulation (Reviewer 1) and Reviewer’s 2 criticism on your approach to interpretation (see attached file) – while revising your manuscript.

Please consider also to provide a map of the region with the sites mentioned in the paper as well as a plan of the tomb and another one showing the tomb in its wider archaeological context (if these are not available, please explain when and where they will be published). References to the radiocarbon dates to the settlement sites seem to be missing (line 160). Finally, please avoid referring to commercial servers such as academia.edu and researchgate in the reference list.

We look forward to receiving your revised manuscript.

Kind regards,

Stefanos Gimatzidis, Ph.D.

Academic Editor

PLOS ONE

Journal Requirements:

Reviewers' comments:

Reviewer's Responses to Questions

**Comments to the Author**

1. Is the manuscript technically sound, and do the data support the conclusions?

Reviewer #1: Yes

Reviewer #2: Yes

2. Has the statistical analysis been performed appropriately and rigorously? 

Reviewer #1: Yes

Reviewer #2: Yes

3. Have the authors made all data underlying the findings in their manuscript fully available?

Reviewer #1: Yes

Reviewer #2: Yes

4. Is the manuscript presented in an intelligible fashion and written in standard English?

Reviewer #1: Yes

Reviewer #2: Yes

5. Review Comments to the Author

Reviewer #1: The article describes an exceptionally important discovery, provides an outstanding analysis and makes insightful interpretation. I greately enjoyed reading it and have only minor queries.

Line 67: who is the 'our' in this sentence? Is this referring to the authors of the manuscript or seeking to make a broader claim about the academic community in general? I found the assumptions on page 3 that all adornments require a pre-exisiting culural code questionable. If that is the case how can any innovation happen? This is particularly important for an object such as that described which is a unique find. Later in the manuscript(pp. 37-38) there is an important discussion about the aesthetic qualities of the necklace, noting the impact they have on the emotional states of the viewers (and continue to have on us today). This can be as important as any conventionalised meaning, and must to some extent precede the emergence of such meanings.

P. 7, line 179: why is the child assumed to be a girl?

PP. 39-40. When discussing wealth, I was surprised there was no comment about the removal of beads from circulation by placing them into burials as a means to control supply and hence maintain their value and status within the living. With the burial of such large numbers of beads this conventional interpretation of controlling supply to maintain value seems to be pertinent.

Reviewer #2: All the technical aspects are perfectly OK.

The title of the article includes: “A reassessment of the techno-symbolic behavior of early farming communities in the Levant”. However, there is no a reassessment of the techno-symbolic behavior of early farming communities. And the geographical term “Levant”, is used here for one site only. This term refers to a large territory: southern Turkey, Cyprus, Syria, Lebanon, Israel, Jorden, the Palestinian Authority and Sinai. Is any of these regions mentioned or discussed? The article does not place the burial in this geographical region. Even nearby Pre-Pottery Neolithic B sites with burials, west of the Jorden, like Jericho, Kefar Hahores, Yiftahel, and others, are not mentioned. This article should have a less pretentious title, such as:An outstanding Neolithic child burial from Baja, Jorden.

6. PLOS authors have the option to publish the peer review history of their article (what does this mean?). If published, this will include your full peer review and any attached files.

Reviewer #1: No

Reviewer #2: No

---

## [Author Response · Author response to Decision Letter 0]

24 May 2023

For a comfortable reading of Response to Reviewers, please check "Response to Reviewers"/section Rebuttal letter at the end of the PDF, after page 65.

We thank the academic editor and the reviewers for their valuable comments and for carefully reading of our manuscript. Their insights have permitted us to improve this article and we hope that this new version elicits a positive response from them.

We have considered all the comments and addressed some of them within the revised version. In this letter, we provide responses (in blue) to each of the raised points. 

Sincerely,

Hala Alarashi 

Corresponding author (on behalf of all co-authors),

To the academic editor

- Both Reviewers 1 and 2 provide substantial comments on content and approaches. In light of these reviews, the manuscript needs some revision. Please consider each of these comments – especially the ones relating to the (economic) implications of the removal of beads from circulation (Reviewer 1) and Reviewer’s 2 criticism on your approach to interpretation (see attached file) – while revising your manuscript.

We considered all the raised issues and provided responses to each one for both reviewers.

- Please consider also to provide a map of the region with the sites mentioned in the paper as well as a plan of the tomb and another one showing the tomb in its wider archaeological context (if these are not available, please explain when and where they will be published). 

We have included a map of the mentioned sites and a plan that illustrates the location of the burial within the broader archaeological context. Regarding the plan of the tomb itself, we believe it is unnecessary for two reasons: 1. it has already been published elsewhere, and we have already cited the reference in which it appeared (Gebel et al. 2019, Fig. 4); 2. the main focus of the paper is on the beads discovered in the burial, which are accompanied by photographs in the main text and drawings depicting their placement within the burial in S1 Appendix. Nonetheless, to provide further clarity, we have added a sentence in line 180-1 as follows: “for a complete description of these events, and the plan of the burial, refer to [58] figs 4, 5, 6 & table 1)".

- References to the radiocarbon dates to the settlement sites seem to be missing (line 160). 

Thank you, reference added. 

- Finally, please avoid referring to commercial servers such as academia.edu and researchgate in the reference list.

All removed and corrected. 

"Reviewer #1

- The article describes an exceptionally important discovery, provides an outstanding analysis and makes insightful interpretation. I greately enjoyed reading it and have only minor queries.

Thank you very much for your interesting comments. Each of them was addressed herein below. 

- Line 67: who is the 'our' in this sentence? Is this referring to the authors of the manuscript or seeking to make a broader claim about the academic community in general? 

"Our" refers to the authors of the manuscript, as the definition was added directly afterwards by citing the definition made by the second author, M. Benz. However, to avoid confusion, we have removed the phrase "our agreed-upon."

- I found the assumptions on page 3 that all adornments require a pre-exisiting culural code questionable. If that is the case how can any innovation happen? This is particularly important for an object such as that described which is a unique find. 

Yes, we agree, it is questionable. This is why we formulated the issue as a question within the same paragraph as following: “Nevertheless, how should one consider outstanding and unique ornaments, innovative creations that do not resemble any of what have existed before? Wouldn't their mere recognition as different justification for inclusion in a code of classification; a first step towards normalization?”. We don’t know if the finds were truly “unique” as it is our vision as archaeologists. Yet, we think that even “uniqueness” may be encoded as such, understood as such. 

- Later in the manuscript (pp. 37-38) there is an important discussion about the aesthetic qualities of the necklace, noting the impact they have on the emotional states of the viewers (and continue to have on us today). This can be as important as any conventionalised meaning, and must to some extent precede the emergence of such meanings.

We absolutely agree on that. The emotional impact experienced by viewers due to the necklace's aesthetics plays a role in formulating the meanings associated with the necklace. We added a sentence to express this idea: “The aesthetic qualities of creations impact our senses and provoke emotions [88,89].Consequently, emotions may play a role in formulating the meanings we attribute to these creations. In other words, the emotional response would be an influential factor that precedes the cultural codes.”

- P. 7, line 179: why is the child assumed to be a girl?

In the text we say, “presumably a girl”. This suggestion is because there are some indications on the mandible (mental protuberance), that it might be a female. However, aDNA analysis failed to confirm this due to poor preservation.

We added this clarification to the sentence and add a reference as following: “A child, presumably a girl (based on the presence of mental protuberance on the mandible) [45] around eight years of age, was buried on the left side in a fetal position (Fig 1D).”

- PP. 39-40. When discussing wealth, I was surprised there was no comment about the removal of beads from circulation by placing them into burials as a means to control supply and hence maintain their value and status within the living. With the burial of such large numbers of beads this conventional interpretation of controlling supply to maintain value seems to be pertinent.

It is indeed a pertinent idea. We heard about practicing withdrawal of beads from life cycle within Papua New Guinee societies. We’ve being searching for bibliography regarding this point but without success. Nevertheless, we discussed more widely this part and included more references: 

“Burying valuable objects such as body ornaments with the deceased may reflect an economic strategy that aims at controlling supplies, thereby at maintaining beads value within living populations. Withdrawing beads from life circulation by hiding or burying them is in fact a universal and intemporal practice, that has been often attested by the discovery of hoards of precious objects [98–100] (usually forgotten by their owners). Some ethnographic examples may illustrate metaphorically the economic interest for beads. For instance, some Krobo people in Ghana believe that specific beads, the old-African powder-glass beads, have spirit and will reproduce in the ground if buried [101]. For other traditional populations such as the Tani and Naga tribes in Northeast India, the sought-after necklaces of glass and stone beads are transmitted over generations as heirloom, and only highly respected individuals are buried with ornaments [102]. Whether the necklace was intended to accomplish an economic objective, to express the esteem to the person of the child, or to provide wealth (or paraphernalia) for the afterlife domain remains an open question. However, we can safely suggest that at least part of the people of Ba`ja had ensured access to prestigious and valued exotic materials, even if they were later transferred into other realms, beyond life. 

 

Reviewer #2

- This article presents an important archaeological discovery, an outstanding necklet associate with an 8-year-old child burial from c. 8500 years ago. Such an elaborate grave good is indeed unique, and above all includes two amber beads, material never found before in the Levant in such an early context. The article should be published by PLOS ONE. In 2019 PLOS ONE published another article on an outstanding burial from that same site. 

Thank you for your careful reading and your insights. Each of them was addressed herein below. 

- The article starts with a general survey on the important of beads in human cultures. Then it describes the site and the burial. Indeed, the excavation conditions were not ideal, so no map of the grave, the skeleton, and the distribution of the grave good is presented in the article. If such map was made, it should be given in publication. These aspects, however, are covered by Figs 1 and 2, together consisted of 15 fields photos. 

- context (if these are not available, please explain when and where they will be published). 

We have included a map of the mentioned sites and a plan that illustrates the location of the burial within the broader archaeological context. Regarding the plan of the tomb itself, we believe it is unnecessary for two reasons: 1. it has already been published elsewhere, and we have already cited the reference in which it appeared (Gabel et al. 2019, Fig. 4); 2. the main focus of the paper is on the beads discovered in the burial, which are accompanied by photographs in the main text and drawings depicting their placement within the burial in S1 Appendix. Nonetheless, to provide further clarity, we have added a sentence in line 180-1 as follows: “for a complete description of these events, and the plan of the burial, refer to [58] figs 4, 5, 6 & table 1)".

- The presentation of the data, beads and the Mother-of-Pearl pendant, is impressive. We are not told about “green beads”, “red beads”, but substantial mineralogical research identify the stones. Other aspects are well investigated as well, using use-wear analysis, and presenting the evidences for bead manufacturing at the site. All these technical studies are impressive. My main problem with this article is on the interpretation level. 

1. The title of the article includes: “A reassessment of the techno-symbolic behavior of early farming communities in the Levant”. However, there is no a reassessment of the techno-symbolic behavior of early farming communities. And the geographical term “Levant”, is used here for one site only. This term refers to a large territory: southern Turkey, Cyprus, Syria, Lebanon, Israel, Jorden, the Palestinian Authority and Sinai. Is any of these regions mentioned or discussed? The article does not place the burial in this geographical region. Even nearby Pre-Pottery Neolithic B sites with burials, west of the Jorden, like Jericho, Kefar Hahores, Yiftahel, and others, are not mentioned. This article should have a less pretentious title, such as:An outstanding Neolithic child burial from Baja, Jorden. 

We regret that the reviewer interpreted the choice of the title as pretentious especially that we sincerely made it to exactly avoid any such “pretentious” understanding of our work. We wanted to avoid terms such as “extraordinary” or “outstanding” because we wished to remain as factual as possible while trying to give an overall insight regarding the implications of this practice “adorning the dead child” within the chrono cultural context of the Levant. We think though that that the second part of the title “reassessment of the techno-symbolic behavior” is clumsy and inappropriate. 

As for the term Levant, we think that we have the right to use it as the site is in the Levant and because it is within the Levantine chrono cultural context that we are discussing this discovery. We did mention other sites in Anatolia, in northern, central and southern Levant. If Cyprus is not mentioned, it is due to its irrelevance for the discussed ideas. As for Sinai, we largely discussed it as it is one of the potential regions of the turquoise provenance. In fact, our study explores many regions in the Levant, and we don’t really understand the criticism of the reviewer here. If some PPN sites are not mentioned, it is because they are not relevant to our study. We made the choice to not state or justify “absence” when it comes to comparisons. We only mentioned the sites that are relevant to the discussed ideas believing that the reader will/would understand why (no mention = nothing to compare or to say about these sites). 

To avoid any “pretentious” interpretation, we finally opted for this title:

Threads of Memory: Reviving the Ornament of a Dead Child at the Neolithic Village of Ba`ja (Jordan)

2. The article raised the question on line 1177-8: “Considering the production and use of the beads raises the question of whether the necklace existed during the lifetime of the child or whether it was created for the death occasion.”. No answer is given here, 

We disagree with the reviewer. In fact, the whole part “Making and sharing memory” discusses step by step our arguments that support the hypothesis of making the necklace for the death occasion: To summarize, the extensive use-wear of certain ornamental elements, namely the “buckle” and the turquoise beads, coupled with the almost pristine condition of the red limestone beads, and the arguments about the ring and bead manufacturing at the site, all indicate that the necklace was composed of both recycled and new beads. Some elements show such heavy use-wear that it seems unlikely they were worn so frequently or for such a prolonged period of time by the child alone. Conversely, the items that appear to have been scarcely used suggest that they were produced specifically for the occasion of death. We then discussed this hypothesis based on the contextual empirical data regarding the complexity of the burial, its construction and other considerations. Therefore, we believe that we have gathered all the arguments to answer this question and suggest, as cautiously as possible, our hypothesis. 

- but in another section, in line 1199:“believed to have been created to endow a highly distinguished 8-year-old child of the community.” The two different possibilities are not discussed at all. There is a need to present the points that support each possibility, and then the authors preference.

It is not clear to us what the reviewer means by opposing these two points as “two different possibilities”. The hypothesis of the creation of the necklace for the death occasion/the necklace was deposited on the chest of the distinguished child? As we said earlier, the first point was largely discussed. As for the second, endowing the dead child with a necklace is for us not a “possibility” but a fact. The child was given an exceptional necklace. Why? We don’t know more that it was because she or he was special due to the elaborated burial and the exceptional features of the ornament. The aim of this article was not to speculate on identities or special status. We aimed at revealing, through this discovery, the symbolic, technological, economic and social implications behind the death ritual and how entangled they are, insofar they insure the cohesion of the community. 

- The authors have the right to have their own interpretation, but in the Southern Levant the body will start rotten fast and the burial ceremony cannot be delayed for days. How much time is needed to plan and create such a sophisticated necklet. 

We would enven suggest that the dry conditions of the southern Levant might even allow for mummification of the body, if no other means were used to preserve it. The use of fire close to the burials is attested for several burials. But this is beyond the scope of the article. 

On the other hand, we didn’t say or mention any correspondence between the moment of death and the moment (time span) of the manufacture of the necklace. We only hypothesized that the necklace was made for the death occasion, although one can imagine many scenarios, for instance, prepared before in the eventuality of death. 

- Or maybe it is not so sophisticated if can be made in a few hours? 

It is very sophisticated and requires time, energy, strings, tools… We have no idea how many people worked together, if the beads to be strung together were partially or completely available, etc. 

3. “a highly distinguished 8-year-old child of the community”. Is it not the highly distinguished status of the family, or the parent?

No not forcingly – in the example of the Dalai Lama – it is a child who is considered the reincarnation. Discussing the status of the family because it is a child who is buried, is what we try to break through. We don’t want to make the simple interpretation: rich burial = rich family. Perhaps he or she was a special child, with extraordinary powers or a strange illness. Who knows? We intend to avoid interpretations for which we have no evidence.

There is no doubt about the child being a distinguished individual. While many other children are buried at the site, several with body ornaments, none of them has the same characteristics of burial and grave goods. 

4. The article suggests that (lines 1203-6) “Its association with a child reaffirms the significant place of children for early farming communities in the Levant and questions their indirect role in stimulating bead production activities, in boosting technologies and aesthetics tastes, especially given that they appear to be the main individuals concerned by body ornamentations in such contexts.” No reference is given to other child burials with beads. Indeed, I do not remember even one continuances Neolithic child burial with beads. So, if there are such burials, they should be cited here. 

Indeed, we did not give references as it is actually a quite known funerary practice and there are many references. We added some examples: “[45,57,132,136,137,148–154]”. 

As far as I can say, the statement: “given that they appear to be the main individuals concerned by body ornamentations in such contexts” is simply not truth! 

It is the opinion of the reviewer that we respect. As we wrote in the conclusion, we are questioning the indirect role of children in boosting the domain of body ornament (the reviewer can consider it as a research perspective). It is our hypothesis and thus far, it seems pertinent to us based on the fact that in many PPNB sites (farming communities) of the Levant and Anatolia (cf. the bibliography above), children seem to be those mainly concerned by body ornaments. 

5. The reconstruction of the necklet is impressive, but it is presented in photo 18 without a scale, a basic requirement in the publication of any archaeological object. This is not just a technical matter, as the size of the complete necklet may indicates if indeed it fits a young child, or better fit an adult. A scale should be added.

It was forgotten. Thank you for noticing. It is done. 

6. The amulet made of Mother-of-Pearl is rather rear, and only two others are known, one of them came from the == burial in Basta (line ===). We are not informed if beads were found at Basta, and what was the age of the buried population. 

We added more information in addition to the previously cited works as following, see lines 739 (approximately) just before table 3: “The ring was found in a vertical position in the grave. Fig. 17 presents it in the same orientation as it was found. The hypothetical reconstruction (details are provided in S3 Appendix) of its shape is based on morphometrical analysis of the preserved parts, and on comparisons made with similar items discovered years ago at Ba`ja (Fig 18) and Basta ([80] fig. 10, 12; [57] fig. 13.3, Hermansen n.d.), although the example from this latter site is much smaller. It is worth noting however that the ring from Basta was likewise discovered with many beads very similar to those of Ba’ja, all associated with an infant (cf. up-cited references). 

7. Two unique Neolithic burials had been uncovered at Baja. Such rich burials ae not known from other Neolithic sites in the Levant. Thus, this is a local phenomenon, maybe reflecting a competition between local families, expressed by extravagant consumption. In any case, these graves cannot help in the reassessment of the techno-symbolic behavior of early farming communities in the Levant.

We disagree with the reviewer. There are many examples from the Levant and Anatolia with children buried with body ornaments. We provided references as requested by the reviewer in the conclusion part. For us, adorning dead children during the Neolithic is not a local phenomenon at all. 

- As written above, the article should be published, but unfounded pretentious assumptions should be removed. 

We respect the opinion of the reviewer and thank her/him for the comments. We regret though that parts of our worked appeared unfounded and pretentious despite all the efforts to, precisely, avoid them. We hope that the revised version fits better his or her expectations.

---

## [Decision Letter · Decision Letter 1]

19 Jun 2023

Threads of memory: Reviving the ornament of a dead child at the Neolithic village of Ba`ja (Jordan)

PONE-D-23-09318R1

Dear Dr. Alarashi,

We’re pleased to inform you that your manuscript has been judged scientifically suitable for publication and will be formally accepted for publication once it meets all outstanding technical requirements.

Kind regards,

Stefanos Gimatzidis, Ph.D.

Academic Editor

PLOS ONE

Additional Editor Comments (optional):

Reviewers' comments:

Reviewer's Responses to Questions

**Comments to the Author**

1. If the authors have adequately addressed your comments raised in a previous round of review and you feel that this manuscript is now acceptable for publication, you may indicate that here to bypass the “Comments to the Author” section, enter your conflict of interest statement in the “Confidential to Editor” section, and submit your "Accept" recommendation.

Reviewer #2: All comments have been addressed

2. Is the manuscript technically sound, and do the data support the conclusions?

Reviewer #2: Yes

3. Has the statistical analysis been performed appropriately and rigorously? 

Reviewer #2: (No Response)

4. Have the authors made all data underlying the findings in their manuscript fully available?

Reviewer #2: Yes

5. Is the manuscript presented in an intelligible fashion and written in standard English?

Reviewer #2: Yes

6. Review Comments to the Author

Reviewer #2: The autobus related to the various comments and corrected the main problems noted, like changing the title of the article, added scale to the original Fig. 18 (now 23) and added references when there were missing. I do not have any additional comments and the article can be published in its revised form.

7. PLOS authors have the option to publish the peer review history of their article (what does this mean?). If published, this will include your full peer review and any attached files.

Reviewer #2: No

---

## [Editor Report · Acceptance letter]

10 Jul 2023

PONE-D-23-09318R1 

Threads of memory: Reviving the ornament of a dead child at the Neolithic village of Ba`ja (Jordan) 

Dear Dr. Alarashi:

I'm pleased to inform you that your manuscript has been deemed suitable for publication in PLOS ONE. Congratulations! Your manuscript is now with our production department. 

Kind regards, 

on behalf of

Dr. Stefanos Gimatzidis 

Academic Editor

PLOS ONE